# The SARS-Coronavirus Infection Cycle: A Survey of Viral Membrane Proteins, Their Functional Interactions and Pathogenesis

**DOI:** 10.3390/ijms22031308

**Published:** 2021-01-28

**Authors:** Nicholas A. Wong, Milton H. Saier

**Affiliations:** Department of Molecular Biology, Division of Biological Sciences, University of California at San Diego, La Jolla, CA 92093-0116, USA

**Keywords:** SARS-CoV-2, Betacoronavirus, Coronaviridae, transmembrane proteins, pathogenesis, inflammation, immunity, vaccines

## Abstract

Severe Acute Respiratory Syndrome Coronavirus-2 (SARS-CoV-2) is a novel epidemic strain of *Betacoronavirus* that is responsible for the current viral pandemic, coronavirus disease 2019 (COVID-19), a global health crisis. Other epidemic *Betacoronaviruses* include the 2003 SARS-CoV-1 and the 2009 Middle East Respiratory Syndrome Coronavirus (MERS-CoV), the genomes of which, particularly that of SARS-CoV-1, are similar to that of the 2019 SARS-CoV-2. In this extensive review, we document the most recent information on Coronavirus proteins, with emphasis on the membrane proteins in the Coronaviridae family. We include information on their structures, functions, and participation in pathogenesis. While the shared proteins among the different coronaviruses may vary in structure and function, they all seem to be multifunctional, a common theme interconnecting these viruses. Many transmembrane proteins encoded within the SARS-CoV-2 genome play important roles in the infection cycle while others have functions yet to be understood. We compare the various structural and nonstructural proteins within the Coronaviridae family to elucidate potential overlaps and parallels in function, focusing primarily on the transmembrane proteins and their influences on host membrane arrangements, secretory pathways, cellular growth inhibition, cell death and immune responses during the viral replication cycle. We also offer bioinformatic analyses of potential viroporin activities of the membrane proteins and their sequence similarities to the Envelope (E) protein. In the last major part of the review, we discuss complement, stimulation of inflammation, and immune evasion/suppression that leads to CoV-derived severe disease and mortality. The overall pathogenesis and disease progression of CoVs is put into perspective by indicating several stages in the resulting infection process in which both host and antiviral therapies could be targeted to block the viral cycle. Lastly, we discuss the development of adaptive immunity against various structural proteins, indicating specific vulnerable regions in the proteins. We discuss current CoV vaccine development approaches with purified proteins, attenuated viruses and DNA vaccines.

## 1. Introduction

The rapid spread of the recently emerging coronavirus disease, COVID-19, caused by the severe acute respiratory syndrome coronavirus-2 (SARS-CoV-2), has given rise to the current pneumonia pandemic, which has a moderate (5%) fatality rate compared to SARS-CoV-1 (10% fatality rate) and Middle East Respiratory Coronavirus (MERS-CoV) (34% fatality rate), due to sepsis/acute respiratory distress syndrome (SARS) [1,2]. Rapid transmission can be hindered by hand washing, distance maintenance between people and the use of masks by infected individuals to trap virus-carrying breath droplets [3]. As of 20 November 2020, there have been over 55 million cases, worldwide, with over 12 million (~22%) being within the USA. The pandemic shows doubling of cases every 5 days and an estimated incubation period of 5 days, both in the USA and world-wide [3]. Well over 1.3 million deaths have resulted, with over 250,000 of them occurring in the USA. The rapid inter-human disease transmission has caught global health professionals by surprise, and as of 20 November 2020, we have no clinically approved anti-viral drug for treatment, and vaccines, recently developed for prevention, are not yet available for distribution [4,5].

The disease arose in the city of Wuhan (Hubei Province, China), and seems to have been caused by the sale of live wild animals as a food source in the Wuhan wholefood market [6]. These animals were thought to be the intermediate carrier of the virus, originating in *Rhinolophus* bats [1]. Although the primary epidemiological factors contributing to the rapid spread of the disease [7] and the immune responses to the virus [8] are reasonably well understood, the animal intermediate carrier has not yet been identified. A greater incidence of the disease in older people, particularly those with co-morbidities such as hypertension and cardiovascular disease, and disproportionate distributions of cases and fatalities among people of different ethnic groups (among African, Hispanic and Native Americans as compared with European Americans) are now well established [9,10].

## 2. Viral Proteins and Their Roles in the Infection Cycle

While the widespread availability of a vaccine is foreseeable, the development of antivirals, targeting specific proteins involved in SARS-CoV-2′s pathogenesis and the infection cycle, may prove to be feasible in a longer time frame. The subsequent progression of disease conditions produced by the 2003 SARS-CoV-1 and the 2020 SARS-CoV-2 include a deadly viral pneumonia and Severe Acute Respiratory Syndrome (SARS), caused in part by the overstimulation of the host’s innate immune system. Patients infected with SARS-CoV-1 or SARS-CoV-2 exhibit similar lung pathological symptoms, again reflecting the similarities between these two viral types. However, additional symptoms affecting other bodily organs have been documented [11,12].

A number of anti-viral agents, known to inhibit or otherwise negatively influence the infectivity or transmission of related viruses, the closest being SARS-CoV-1, have been considered, and some of these have even been tested, either in vitro or in vivo. For example, micronutrients such as vitamins C, D and E have been suggested to be of value for prophylaxis or treatment [7]. Controversial therapies include treatment with inhibitors of the angiotensin-converting enzyme, ACE, the receptor for SARS-CoV-2 [13], non-steroidal and steroidal anti-inflammatory drugs such as ibuprofen and corticosteroids, and various Chinese herbal medicines. These have all been considered and recommended in published articles [7,14,15]. However, several antiviral agents, effective against other viruses, are being tested, and in some cases, they are being used in clinical trials. One of the most promising drugs is remdesivir and its derivatives, broad spectrum antiviral drugs; current limited evidence suggests that they have been effective as a therapeutic option in combating COVID-19 [16,17]. Chloroquine (CG) and hydroxychloroquine (HCG), anti-malarial drugs, have been used, but caution is required because of the toxicity of these drugs and because higher doses seem to be required to combat the virus than are used for malaria. At present, there is insufficient evidence to suggest that CQ/HCQ are safe and effective treatments for COVID-19, and most of the evidence suggests that at safe doses, they are not effective [18,19].

Potential drug targets of SARS-CoV-2 include several viral structural and non-structural proteins. The effective development of novel drugs specifically for this virus depends on an understanding of the structures and functions of the constituent proteins [20]. The genome sequences of several *Betacoronaviruses* (β-CoVs) have been available for several years, and that of the new CoV-2 has been available almost since the initial outbreak in the city of Wuhan [21]. A typical betacoronavirus genome includes at least six genes (open reading frames; ORFs), some or all of which can be post-transcriptionally/translationally proteolytically cleaved into smaller proteins or peptides with importance to the viral infection cycle, and these may prove to be suitable drug targets [22]. These proteins include the structural Spike (S), Envelope (E), Membrane (M), Nucleocapsid (N), Hemagglutinin esterase (HE) and Helicase (H) proteins, and nonstructural proteins (nsp) which include Proteases, papain-like proteases (PLP or PL^pro^) and 3C-like protease (3CL^pro^), and Replicase proteins [22]. The Spike (S) protein, found as protrusions on the surface of the viral particle, includes the fusion peptide while the envelope (E) protein has viroporin activity, important for normal completion of the viral infection cycle. The nucleocapsid (N) protein forms a nucleoprotein with the single (+) stranded RNA genome, and this nucleoprotein complex is maintained within the capsid comprised in part of the matrix (M) protein. The protease proteins, which lie in ORF1a/ORF1ab, have been a target of anti-viral action by lopinavir/ritonavir, with and without arbidol, an anti-envelope viral indole derivative [16,23]. Four main structural proteins are encoded by ORFs 10 and 11 near the 3′ terminus of the viral genome (Figure 1) [21]. Among the potential targets of drug action are E-protein viroporins [24] and ‘Spike’ membrane fusion proteins [25]. In this review, we discuss several of the SARS-CoV-2 proteins but focus on these two proteins as potential targets of novel potential drugs to combat COVID-19, SARS and MERS.

## 3. The CoV Replication Cycle (from Entry to Exit)

The coronaviruses comprise a family of enveloped RNA viruses with positive sense strand genomes, meaning that expression of most genes, with the exception of the late structural genes, are transcribed directly from the genome. The first obstacle of any virus is entry into its host cell, utilizing common and often highly expressed viral receptors present on the host membranes. These receptors not only play a role in adherence to the host surface but also facilitate membrane fusion [26]. Attachment to host surface proteins often leads to a chain reaction that activates entry mechanisms in either the host, the virus, or both [27]. At some point during or after viral entry into the host cell, the virus may release its genome to begin hijacking the cellular machinery, allowing replication of its genome with the eventual production of viral products (Figure 2). These receptors may vary widely amongst viruses; they may be carbohydrates as is the case for influenza virus which uses sialic acid in respiratory epithelial cells, or they may be specific for Clusters of Differentiation (CD) ligands on immune cells. For example, HIV targets CD4 and CCR5/CXCR4 of human T-cells. Viruses may also utilize coreceptors that do not activate entry mechanisms but increase their affinity for their host to improve the probability of eventually engaging the correct viral receptor for entry.

For enveloped viruses, entry into the host involves fusion of the host membrane with its own viral envelope to release its genome into the cell (Figure 2). All human infecting coronaviruses are capable of infecting the respiratory tract, and the three epidemic strains of β coronaviruses, SARS-CoV-1, SARS-CoV-2 and MERS, take advantage of prevalent ectopeptidases, surface glycoproteins found not only in respiratory organs, but also in epithelial cells of the gastrointestinal (GI) tract and excretory system as well as immune cells [28,29]. Dipeptidyl peptidase-4 (DPP4) is utilized by MERS, and both SARS-CoV-1 and SARS-CoV-2 use human angiotensin-converting enzyme 2 (ACE2) [26,30,31,32]. ACE2 is commonly found throughout human epithelial tissues including the oral and nasal mucosa, nasopharynx, lungs, stomach, small intestine, colon, lymph nodes, thymus, bone marrow, spleen, liver, kidney and even the brain which sometimes leads to systemic infection in late pathogenesis of Covid diseases [29]. For both SARS-CoV-1 and SARS-CoV-2, The S1 subunit of the Spike (S) protein engages ACE2 and primes activation of the S fusion core complex by host proteases. The S protein fusion complex of β-CoVs is promiscuous and can be cleaved by a variety of proteases such as furin, trypsins, trypsin-like proteases, cathepsins, PC1, transmembrane serine protease-2 (TMPRSS-2), TMPRSS-4, type II transmembrane serine protease (TTSP) matriptase, and human airway trypsin-like (HAT) protease [32,33]. Membrane fusion, and release of the viral genome into the host cell may occur through a non-endosomal pathway at the membrane surface of the cell, although this is a minor pathway for entry in SARS-CoV-1 and MERS [34]. Studies have confirmed that SARS-CoV-1 and MERS prefer to enter the host cell through endocytic pathways, activating the fusion complex and releasing their genomes after maturation of the endosome following initial entry, similar to the endosomal pH-dependence of influenza viral fusion complexes [35]. However, it is the availability of these host proteases, present at extracellular binding sites, that dictates if the CoV will fuse its membrane at the surface, or after endocytosis, which may explain the varying infection dynamics amongst different cell types [32]. For instance, colocalization of ACE-2 and TMPRSS-2, possibly on lipid rafts of cell surfaces, may optimize cell to cell entry of SARS-CoVs and syncitia formation in respiratory epithelial cells [36].

Induction of endocytic pathways is a common theme among enveloped viruses, often utilizing commonly characterized methods such as clathrin- or caveolae-coated pits, pinocytosis, or macro-pinocytosis [37]. Drugs inhibiting endocytic pathways have proven to be effective antivirals for various viruses such as influenza virus, simian virus, herpes simplex virus, and coronaviruses [35,38,39,40]. Human infecting coronaviruses have been shown to use clathrin- and caveolae-mediated endocytosis; however, alternative SARS-CoV-1 entry mechanisms, independent of clathrin and caveolae, have also been confirmed, distinguishing SARS-CoV-1 from the other coronaviruses and enveloped viruses in general [34,41,42,43].

Clinically available chlorpromazine (CPZ), a clathrin inhibiting drug, significantly reduced MERS entry and infection rates, but not those of SARS-CoV-1 [34,44]. Additionally, sequestering cholesterol lipid raft-dependent caveolae-mediated endocytosis with filipin or nystatin did not disrupt SARS-CoV-1 entry [34]. The lack of reliance on clathrin and caveolae for cell entry of SARS-CoV-1 is similar to influenza viral entry, which can also infect cells via clathrin- and caveolin-independent endocytic pathways [35]. Interestingly, Ou et al. reported that SARS-CoV-2 endosomal entry can be inhibited using apilimod, a potent phosphoinositide 3-kinase (PIKfyve) inhibitor. PIKfyve is the only mammalian producer of phosphatidylinositol-3,5-bisphosphate (PI(3,5)P_2_) in early endosomes, and it regulates endosomal sorting [32,45]. Although Ou et al. reported no in vitro cytotoxicity of apilimod at any of the tested concentrations on HEK293 and HeLa cell lines, despite its low abundance in endosomes, the PIKfyve-PI(3,5)P_2_ complex is crucially important to cellular homeostasis and signal pathways [45,46]. Disruption of this complex has been linked to a variety of neurological diseases and neurodegeneration in humans with links to Alzheimer’s disease and amyotrophic lateral sclerosis (ALS) [45,46].

Downstream of PI(3,5)P_2_ is the NAADP-PI(3,5)P_2_-activated cation-sensitive Two Pore Calcium Channels (TPC) [47]. TPC1 (TCID 1.A.1.11.25) and TPC2 (TCID: 1.A.1.11.19) have been shown to play roles in calcium signaling in endosomes and facilitate the fusion of acidic protease-rich lysosomes with endosomes to form late endo-lysosomes, providing a versatile array of cell type-specific responses [48,49,50]. Generally, both TPC1 and TPC2, but primarily TPC2, efflux calcium from endosomes into the cytosol of the cell to signal endo-lysosomal fusion, with TPC2 localizing to both early and late endosomes as well as lysosomes [51,52]. Mature endo-lysosomes may continue to use TPC2-mediated calcium signaling to regulate endo-lysosome interactions with the ribosomal rich ER membrane, a region of importance for enveloped RNA viruses [51].

Interest in TPC2 has grown after confirmation of its role in NAADP-induced calcium signaling for endosomal sorting and maturation, relevant to virus entry. Inhibition of TPC2 has shown positive results in preventing endosomal viral entry into cells for both enveloped and nonenveloped viruses. The enveloped virulent viruses, Ebola [53], MERS [54], SARS-CoV-2 [32], and henipaviruses [55] all show weakened or abrogated abilities to release viral contents into the host cell cytosol after either deleting or inhibiting TPC2, indicating that TPC2 is a necessary and common entry participant for various viruses. Disruption of TPC2 activity also blocks translocation of virus-containing endosomes through the endocytic pathway in MERS infected cells [54]. Unfortunately, none of the studies pinpointed a more complete reason as to why TPC2 is crucial for viral entry, but an understanding of the requirements of viral fusion proteins may provide insight.

Another overlap among the mentioned viruses is the dependency on the host endolysosomal protease, cathepsin L, which cleaves and activates the various viral fusion proteins [56]. While other proteases may be used by various viruses, SARS-CoV-2 showed a specific dependency on cathepsin L; deletion of this lysosomal protease drastically decreased viral release [32]. TPC2′s calcium signaling could be a crucial step in (1) endo-lysosome formation, (2) incorporation of lysosomal proteases, (3) acidification of the virus-containing endosome as a prelude to membrane fusion/viral release, and (4) navigation to the ER-golgi membrane [51] where membrane-associated protein synthesis occurs. Indeed, TPC2 inhibition with fangchinoline inhibited endosomal furin protease activity and mobility of endosomal structures [54]. It remains elusive if TPC2-mediated signaling is required for cathepsin L protease activity. To the best of our knowledge, no such study exists as of yet connecting the full nature of endosomal maturation pathways, protease activity and viral membrane fusion. 

## 4. Coronavirus Genome Structure, Replication and Expression

The coronavirus genomes are the largest of any positive stranded RNA viruses, ranging from 26 to 32 kb [57]. Each of these genomes is a single continuous strand of RNA that mimics host mRNAs with the existence of a 5′ cap and poly A tail [58]. Coronaviruses share a common structure, but in the interest of the epidemic strains of β-CoVs, we will primarily discuss the genomes of MERS, SARS-CoV-1, SARS-CoV-2 and other analogous β-CoVs. The 5′ end of the genome contains ORF1a and ORF1ab which contain the early nonstructural genes that compose the CoV replication-transcription complex (RTC) (Figure 1).

While the genes in the RTC region are directly expressed, replication and transcription of the genome is more complicated. Like other RNA viruses, Coronaviruses encode their own RNA-dependent RNA polymerases (RdRp) for genome and gene amplification. Both genome replication and expression of sub-genomic RNAs towards the 3′ end of the genome require reverse transcription and production of negative strand mRNA intermediates. Further complicating amplification and expression of late genes is the presence of transcription-regulating sequences (TRSs) that exist at the beginning of each structural or accessory gene [59]. These TRSs are composed of clusters of weak affinity oligonucleotides; for instance, SARS-CoV-1 TRSs contain the core sequence (5′-AAACGAAC-3′) [60]. These unique TRSs act as RdRp pause sites, giving the viral replication complex opportunity to dissociate and randomly walk around the genomic template until it finds another TRS to resume transcription. Transcription or random recombination of these sub-genomic RNA strands does not stop until the RdRp reaches a final TRS at the leader sequence on the 5′ end of the genome [59]. Alternatively, the RdRp can continuously transcribe through the genome, creating complete negative strand templates for full genome replication. This novel ability to discontinuously transcribe negative strand templates and recombine transcripts can provide insight into how CoVs are so readily able to mutate and recombine into novel strains. Coinfection of a single cell by two different CoVs can experience an RdRp jump from the TRS of one genome to that of another CoV genome, potentially creating a recombined genomic transcript [61]. Altogether, negative strand RNA templates comprise only up to 1–2% of the total viral RNA products in an infected cell, implying that only a few negative strand templates are required for amplification of full genomes and sub-genomic late genes [59,62]. All negative strand templates must then be reverse transcribed, again to create positive strand mRNA templates for protein synthesis or production of viable genomes for packaging.

Like other positive sense single stranded RNA viruses, early expression genes are transcribed and translated directly off its genome. These early genes exist in two nested open reading frames, ORF1a and ORF1ab, and include the nsps (Figure 1). These proteins are translated by cellular ribosomes into polyproteins, ORF1a polyprotein (pp1a) and ORF1ab polyprotein (pp1ab) [63]. pp1a is cleaved into 10-11 nsps, and pp1ab is cleaved into as many as 16 mature viral nsps that participate in immediate catalysis of viral RNA synthesis (Table 1) [64]. Cleavage of these polyproteins is accomplished by virally derived papain-like proteases PL1^pro^ and PL2^pro^ in nsp3 (but only PL1^pro^ designated as PL^pro^ in SARS-CoV) [65] and 3CL^pro^ in nsp5 [66]. nsp3 cleaves off itself, nsp1 and nsp2 from pp1a or pp1ab while 3CL^pro^ cleaves itself from nsp4 and nsp6, and it additionally cleaves nsp7-16 [67]. Expression of both pp1a and pp1ab is made possible by a heptameric ‘slippery sequence’ (5′-UUUAAAC-3′) and a ‘pseudoknot’ that exists at the end of ORF1a before ORF1ab. In SARS-CoV-1, and likely other betacoronaviruses, the pseudoknot is composed of 3 highly conserved, thermodynamically stable stem loops (free energy ~ −21.12 kcal/mol). These stem loops are digestible by dsRNAses, and disruption of any of these stem loops reduces frame shifting and expression of pp1ab by up to 94% in vero E6 cells [68]. The pseudoknot occasionally restricts the ribosome from proceeding farther than the ORF1a stop codon, halting expression at this point and allowing for the ribosome to ‘slide’ on the slippery sequence. This ribosomal sliding may shift the reading frame backwards by 1 nucleotide from ORF1a to ORF1ab, extending translation to express pp1ab [59]. However, most of the time, the ribosome is able to melt the pseudoknot, progressing through the slippery sequence uninterrupted until it reaches the ORF1a stop codon, producing more pp1a than pp1ab [68,69].

ORF1a encodes nsp1–11, which include several membrane spanning proteins associated with viral replication. Within ORF1ab are the genes for the important nsp12–16 which all directly participate in viral RNA replication [63]. Collectively, the nsps derived from ORF1a and ORF1ab are called the RTC [62]. The important nsp12 functions as the viral RdRp for reverse transcription and is associated with nascent viral RNA synthesis [62]. To emulate host mRNAs, provide stability, ensure proper translation, and avoid degradation by host nucleases, the viral genomic and nascent RNAs are all capped and methylated at the 5′ end via nsp14 N7 cap methyltransferase [84]. With expression of the RTC, the virus is equipped with the proper arsenal to hijack the cell, requisitioning and modifying important host organelles and the machinery for its own replication.

## 5. CoV Replication Organelle Formation: A Dance of Membrane Rearrangements

In order to review the nature of coronavirus membrane reorganizations, it is important to note that all nidovirales rearrange host membranes in similar manners, thanks to nsps derived from pp1a and pp1ab. In this section, we present collected information from various studies of early membrane rearrangements induced by members of the Nidovirales (Coronaviruses and Arteviruses) with an emphasis on β-CoVs (SARS-CoV-1/2, MERS-CoV and MHV). Throughout the replication cycle of Nidovirales viruses, dramatic rearrangements of host membranes are utilized by the viruses for replication, protein expression, assembly and exit. Virally induced zippered ER, production of double membrane vesicles (DMVs), convoluted membranes (CMs), and later Vesicle Packets (VPs), Double Membrane Spherules (DMSs), the occasional membrane whorls (MWs), and giant vesiculations (GVs) are hallmarks of coronavirus and other nidovirales infections. Other more elusive structures include cubic membrane structures (CMSs) and tubular bodies (TBs) [84]. All membrane rearrangements are visible through electron microscopy (EM) and electron tomography (ET) and occupy generous portions of the cell space. While minute details will obviously cause slight differentiations in the dynamics of early infection, the nsps responsible for membrane rearrangements are astoundingly similar amongst the Nidovirales, allowing for a comparative analysis within the virus order. See Table 2 for comparisons between important membrane rearrangements amongst different Coronaviruses.

In all (+) stranded RNA viruses, viral-induced reorganization of cellular membranes has been documented and associated with viral RNA synthesis [85]. Benefits of associating viral RNA synthesis with modified membranes include more efficient catalysis of viral products with association of macromolecule-rich membranes such as the ER and Golgi. Viruses may reorganize these organelle membranes into ‘viral organelles’ to separate specific parts of viral replication stages and allow induction and confinement of RNA synthesis and dsRNA intermediates to microenvironments shielded from host innate antivirals [86]. Viral organelles specific for replication and nucleotide synthesis are appropriately called replication organelles (ROs). Nidovirales, and more specifically, the subfamilies Coronaviruses and their related subfamily Arteriviruses appear to have a novel rearrangement of host membranes not seen in any other (+) sense RNA viruses [85,87]. In β-CoVs, early expression and cleavage of nsp2–6 TM proteins participate in membrane rearrangement of the ER into DMVs which are completely sealed from the cytosol [MERS, SARS, MHV, HCoV-229E, PEDV, IBV], with cleavage of nsp3 and nsp4 being crucial for SARS-CoV-1 and MERS. Plasmid expression of only MHV or MERS-CoV-1 nsp3 and nsp4 was sufficient to replicate clustered DMV membrane rearrangements reminiscent of MERS-infected cells [88,89]. Similarly, SARS-CoV-1 nsp3 and nsp4 were sufficient to induce phenotypic membrane rearrangements reminiscent of SARS-CoV-1 infected cells [89], but nsp6 may additionally promote these rearrangements [57]. Together, these findings imply that late membrane-bound structural proteins are not required to modify host membranes in the early infection, and this process is primarily associated with the RTC.

In the following text, we attempt to provide an approximate timeline of CoV RO formations with an emphasis on β-CoVs. Nearly immediately after the early synthesis and expression of RTC RNAs, viral replication complexes begin to form within cytoplasmic membranes. Abundant transmembrane nsps begin to accumulate in the membranes of the ER and induce unusual modifications, localizing in the perinuclear region of the cell. With β-CoVs, as early as 1–2 hpi, isolated DMVs can be seen forming in the cytoplasm with a slight proximity to the ER, which may exhibit a slight zippering shape [90,91]. Low levels of viral RNA synthesis are also detectable in MERS, SARS, MHV, and IBV [62]. Shortly after, in β-CoVs, large CMs (0.2–2 µm) and reticular inclusions form with connections to DMVs and the ER [90,91]. By 4 hpi, DMVs have drastically increased in amount and cluster throughout the cell with increasing localization in the perinuclear space [90,91]. At some point, the golgi membrane is also incorporated in the DMV-CM-ER network [91,92]. By 5–7 hpi, newly assembled and budding virions appear in the Golgi cisternae as do virion-containing golgi-derived LCVCs [90,91,93]. α-, β-, δ- and γ-CoV ER morphologies adopt a zipper form [92,93,94]. γ- and δ-CoVs also contain DMSs sprouting from the ER with small ‘neck like’ openings to the cytosol [93,94]. At 7 hpi and beyond, there can be as many as 200–300 DMVs clustered within the cells infected with SARS [90]. After this point in time, the outer membranes of several DMVs have fused together to create an interconnected system of single membrane vesicles contained within one communal outer membrane forming large VPs. At least 95% of DMVs have one or multiple thin ‘neck like’ connections (~8 nm) to one or several other DMVs, CM or the ER, implying that late stage modified membranes form one large uninterrupted network with the ERGIC membranes [90,91]. Late infection membrane rearrangements in α- and β-CoVs may also include DMSs embedded in CMs and have interconnections with CMs, CM-ER, and ER membranes, although the exact time frame when these are first formed in β-CoVs is not clear [92,93,95]. This network is appropriately called the reticulovesicular network (RVN). [64]. Finally, at ≥10 hpi, assembly of new virions can be seen budding into and from the heavily modified ERGIC lumen, and the virus has completely repurposed the cell for viral replication [90,92].

There can be up to 1000 vesiculations in a single SARS-infected cell [90], and separate VPs fuse together to form GVPs which may be interconnected with LVCVs and Golgi membranes containing both vesicles and significant numbers of budding complete virions [90,95]. In all CoVs, at every recorded stage, no DMVs exhibit openings to the cytosol [64,90,91,92,93,94,95,96]. DMVs close to the ER exhibit tightly apposed inner and outer membranes, but perinuclear residing DMVs have looser organization between the two membranes [90].

While the patterns of membrane rearrangements are generally common amongst all coronaviruses studied thus far, slight differences do exist among the families. γ-CoV IBV and δ-CoV PDCoV did not show visible membrane rearrangements until after 6 hpi, but the phenotypes were remarkably similar to those of the observed β-CoVs and remained consistent through to 24 hpi [92,93,94]. Another stark difference between β-CoVs and γ-CoVs is the organization of CMs and DMVs. In β-CoVs, DMVs likely form first and cluster in a net of CMs [90,91], while in γ-CoVs, zippered ERs and DMSs are the primary structures, and DMVs tend to appear as free-floating vesicles away from the ER [94]. α-CoVs, HCoV-229E and PEDV, have membrane rearrangement patterns similar to those of β-CoVs, but they take between 24–60 hpi to develop [92,95]. PEDV also exhibits unique late infection membrane rearrangements called endoplasmic reticular bodies (ERB) which occur in a minority of cells and virion-positive endolysosomal compartments [95].

As mentioned before, β-CoVs require nsp3 and nps4 for modification of cell membranes. Exactly how these nsps are capable of modifying the ER membrane to produce zippering and vesiculations remains uncertain. Disruption of nsp3 and nsp4 expression greatly inhibits RVN formation and viral genomic replication [97,98]. Co-expression of nsp3, nsp4 [88,89] and nsp6 (for SARS) [57] induces clustered DMVs and disorganized double membraned CM-like structures sprouting from the ER. It is thus proposed that accessory structures other than DMVs are induced by other viral proteins or viral replication.

Initially, it was thought that host vesicular or secretory pathways might associate with these nsps to induce membrane changes. Autophagosomes and endo-lysosomes have been known to be induced by coronavirus infections and appear during mid-late infection, but they are not required for viral replication [95,99]. Additionally, during mid infection, virion positive Golgi born LVCVs form in conjunction with ERGIC and DMV constructs [91]. However, no matter which CoV, there exists a high statistical correlation between the abundance of different membrane rearrangements. In MHV, the abundance of DMVs with CMs in a single cell is comparable to the abundance between DMVs and DMSs in an IBV infected cell [100]. These findings provide evidence that membrane rearrangements for a given CoV are highly related to each other in the formation of an RVN. Perhaps a dominating structure in a CoV may be able to compensate in function for another missing membrane structure.

Picornavirus utilizes some parts of the secretory pathway for membrane reorganization (DMVs) and replication [101]. Treatment of infected cells with brefeldin A, an antiviral drug that inhibits secretory protein transport from the ER to Golgi, suppresses host secretory pathways completely and prevents picornavirus replication, but it only partially inhibits coronavirus replication [64]. In fact, upon suppression of native host secretory pathways, early induction of DMVs was hindered merely to 20% of normal activity, and late-stage infection exhibited a similar phenotype [64]. Interestingly, this experiment revealed that host ER secretory protein Sec61α (TCID: 3.A.5.9.1), a transport protein subunit that anchors ribosomes to ER membranes and shuttles unfolded polypeptides into the lumen of the ER, was redistributed to RVNs upon SARS infection. However, of interest, Brefeldin A treatment of SARS-infected cells (1–7 hpi) accelerated RVN formation, while slightly inhibiting the expression of nsp3 and N protein causing small differences in RVN morphology. DMV formation seemed to be accelerated, and aggregation into LVCVs occurred during mid infection (7 hpi) as opposed to late infection times. Additionally, the luminal space between inner and outer membranes of DMVs appeared more open, but not on the sides facing CMs. Intracellular virions could still form, but brefeldin A treatment prevented secretion of these particles, due to downstream inhibition of excretory pathways necessary for virion budding [64]. Hence, suppression of secretory pathways slows early CoV infection and lowers virion productivity, but it does not stop RNA synthesis and virion production altogether.

Protein disulfide-isomerase (PDI), another luminal ER-associated protein that participates in the formation of disulfide bridges of unfolded polypeptides, migrated to MHV- and SARS-induced RVNs and also partially localized with nsp3 [64,102]. Interestingly, when nsp3 or nsp4 is singly expressed within a cell, it localizes with PDI, but not when they are co-expressed [88]. Unfortunately, what the relationships between CoV RVNs and host PDIs are remains elusive. Any antiviral responses involving suppression of host ER protein interactions with nsps must be further investigated [102]. Autophagosomes have also been proposed to be involved in membrane rearrangements, but several studies seem to have argued against this hypothesis, as deletion of autophagy-related-genes (encoding ATGs) did not prevent DVM formation, despite a SARS nsp6 association with the ATG pathways [103,104,105]. Additionally, microtubules are not required, despite LC3 decorating the outer membranes of DMVs [85].

This departure from common viral replicase themes in early membrane rearrangements has brought Nidovirales research to the currently supported hypothesis that the nsps themselves are sufficient to induce these dramatic membrane rearrangements. Associated host pathways discussed above merely support mechanisms that promote membrane rearrangement without being required. Bioinformatic analyses of the TM domains as well as NMR and X-ray crystallography analyses of these nsps, support the theory that simply the shapes and protein-protein interactions amongst ER spanning nsps is enough to scaffold membrane rearrangements. Both nsp3 and nsp4 span the membrane multiple times and induce the ER early membrane pairing necessary for RVN formation. Intriguingly, nsp3 and nsp4 of β-CoVs and their equivalents in other CoVs retain only modest primary sequence homology, in spite of strong structural similarity, specifically, in the proximal location of the TMDs and large luminal loops [106]. This could explain why early infection membranes are characterized by a zippered ER with minimal vesiculation in some CoVs. Over time, as RNA synthesis increases, expression of pp1a and pp1ab allows for sufficient accumulation of membrane spanning nsps in the ER, causing dramatic vesiculation and reconstruction of host membranes. Differences in dominating structures may be attributed to small differences in integral membrane nsp topologies, or to other infection dynamics. Lone expression of nsp3 or nsp4 individually only localizes the protein to the ER, but co-expression of nsp3 and nsp4, or polyprotein nsp3–6, gave rise to perinuclear localization of these nsps and formation of clustered DMVs [57,88]. Structural inspection of these nsps revealed that a truncated nsp3, where the deletion spans from the first TMS to the C-terminus, can still localize with nsp4 to perinuclear regions [88]. Even more interesting is a possible direct involvement of nsp3 and nsp4 in viral replication in addition to RO formation.

## 6. Structures of nsp3 and nsp4

Initially, pp1a and pp1ab are cleaved by the PL1^pro^ or PL2^pro^ papaine-like proteases that are encoded in the region adjacent to nsp3. In SARS and MERS-CoVs, only the PL2^pro^ protease is present, but it performs the same function and cleaves the pp1a or pp1ab into the nsp1, nsp2, and nsp3-polyproteins. Further proteolysis of the nsp3-polyprotein is fulfilled by nsp5 to generate the individual nsp3, and remaining nsp proteins. Also, to the N-terminal side of nsp3 is a ubiquitin-like domain 1 (Ubl1) which, while retaining poor primary sequence conservation amongst CoVs, has well conserved secondary folding and is present in all CoVs [98]. Ubl1 is associated with ssRNA, interacts with the N protein, and is essential for viral RNA synthesis [106]. Ubl1 weakly binds to N, but the exact regions that associate with each other may vary among CoVs [98]. However, deletion of Ubl1 in MHV prevents viral replication [98]. It is possible that Ubl1 serves as a dock for viral genomes to early RTCs, bringing N and the nascent viral genomes close together for packaging. This would suggest that nsp3 not only facilitates the remodeling of host membranes, but also serves as an active protein in genome packaging. Additionally, the Ubl1 of SARS-CoV-1 might disrupt Ras regulated cell-cycle progression. The Ubl1 is similar to the Ras-interacting domain of the Ral guanine nucleotide dissociation stimulator (RalGDS). Since SARS and MHV infections are known to induce cellular arrest in the G_0_/G_1_ phase, it could be that nsp3/Ubl1 disrupts the interaction between Ras and its downstream effectors [98].

Following the Ubl1 domain are the macrodomains (Mac1, Mac2 and Mac3), with Mac2–3 forming a portion of what was once thought of as a SARS Unique Domain (SUD). It was thought the SUD was unique to SARS due to the high variability in CoV genomes, but secondary structure analysis has revealed strong structural similarity in these regions amongst other CoVs, and Mac2-3 may not be unique to SARS [106]. The Mac domains are all similar in structure and are hypothesized to be gene duplicates of Mac1. However, only Mac3 was shown to be necessary for SARS replication in a cDNA study [107]. Following Mac3 on the C-terminal end is Domain Preceding Ubl2 and PL2^pro^ (DPUP), which forms antiparallel β-sheets. The Mac2-Mac3-DPUP complex has an affinity for nucleic acids and binds to RNA. Specifically, Mac3 was shown to bind oligo(G) [107] and oligo(A) [98]. Thus, Mac3 may bind to the poly(A) tails of RNA molecules present in viral genomic, sub-genomic and host RNAs. Also present in Mac3, is a unique antiparallel β sheet that exists in MHV [108]. Following the DPUP and Ubl2 domains is PL2^pro^, previously mentioned to cleave the nascent viral polyprotein into nsp1/2, nsp2/3, and nsp3/4 [109]. In addition to autocleavage activity, PL2^pro^ has deubiquitinating and deISGylating activities, which may participate in suppression and evasion of intracellular innate immune pathways [98,109]. Note: ISGylation is the process of IFN-induced gene ISG15 ubiquitin-like protein associating with targets (ISG15 targets) [110]. The SARS-CoV-1 Mac2-3 and PL2^pro^ domains together may elicit innate immune suppression by competitively binding to host E3 ubiquitin ligase RCHY1, leading to down-regulation of antiviral pro-apoptotic transcription factor p53 [111,112]. In contrast to the enzymatic nature of the N-terminal side of nsp3, the C-terminal one third end of the nsp3 protein (nsp3C) interacts with nsp4 and other nsps (e.g., nsp8) [57,88,97,98]. The specific interaction of nsp3 with nsp4 may induce the hallmark membrane curvature in the ER.

The nsp4 protein has four TMSs, one large luminal loop between the 1st and 2nd TMSs, and a small luminal loop between the 3rd and 4th TMSs. The 3rd and 4th TMSs are dispensable for SARS and MHV nsp4, but deletion of TMSs 2–4 affects localization with nsp3. Moreover, deletion of either TMS 1 + the large loop, or TMSs 2–4 + this large loop, completely prevents localization with nsp3 and nsp3c [88]. Changing the luminal loop of nsp4 of one CoV for another also prevents localization with nsp3, suggesting that the exact structure of the N-terminal large luminal loop is specific per CoV nsp3c binding site [88]. Within the large luminal loop resides 4–10, cysteine residues as well as glycosylation sites are conserved amongst CoVs [57,88]. Deletion of glycosylated regions produces aberrant DMVs with large luminal spaces and increased levels of CMs [57]. Replacement of the cysteine residues results in low localization with nsp3, suggesting possible disulfide bridge formation amongst nsps during membrane pairing [88], as well as a possible reliance on PDI and Sec61α. In SARS, the region responsible for localizing with nsp3C are within the regions 112–164 aas and 220–234 aas. Deleting those two regions, or only the specific residues H120 and F121 within nsp4, prevents localization with nsp3C, replicon formation and thus, viral replication, but the phenotype can be rescued when wild type nsp4 is reintroduced via an encoding cDNA [97].

In SARS-infected cells, nsp6 is required to induce an RVN phenotype akin to wildtype. Without nsp6, DMVs migrate farther away from the RVN. Nsp6 may break up elongated stretches of membrane pairings, causing DMV vesiculation to cluster near CMs, causing RVN formation for SARS. The existence of polyprotein nsp4–6 following nsp3 cleavage may be critical for efficient SARS DMV formation [57]. Overall, nsp3, nsp4 and nsp6 are essential for normal SARS replication. In the pursuit of antivirals, inhibition of the papain-like proteases that cleave the early polyproteins would be an excellent upstream target to block early viral replication.

## 7. The Replicon Conundrum and a Putative Nucleopore

Based on the themes of other (+) sense RNA viral replicases, it would be expected that coronaviruses may modify membranes for similar reasons as for the existing canonical replication themes of many RNA viruses, using DMVs as RNA factories. In other RNA viruses, the existence of dsRNAs is often a marker for nascent viral RNA synthesis since dsRNA molecules are a necessary intermediate. Labelling for dsRNA in CoVs revealed that their presence within DMVs could be observed throughout infection, with the signal growing stronger over time. Following the logic of other RNA viruses, perhaps DMVs might provide an encapsulated environment free of host innate immune mechanisms to protect viral RNA synthesis. Utilizing membrane rearrangements sourced from the ER would also benefit CoVs as their proteomes contain many essential structural and nsp TMD-containing proteins. However, in all previously recorded CoVs and Arteviruses, the DMVs are sealed from the cytosol, and nsps delocalize from DMVs during mid to late infection [90,92]. Spatially, DMVs migrate away from the ER-CM constructs and cluster in the perinuclear regions forming the VPs in conjunction with modified Golgi and LVCVs during late infection. Electron micrographs of some VPs and virion budding LVCVs documented that the connected membranes seem to compartmentalize, keeping former DMVs on one side while mature and budding virions form on the other [90,92].

Studies using either BrU or Click chemistry and labelled RdRp and an nsp (nsp8 or nsp12) to detect nascent RNA synthesis revealed that CoV viral synthesis first localizes with dsRNAs and TM nsps, but migrates to perinuclear regions later in infection, eventually spreading throughout the cell [62]. On the other hand, nsps primarily accumulate in the ER and CMs [62,64]. Additionally, dsRNAs did not incorporate BrU or clickU, implying that these intermediates were catalytically inactive and did not participate in RNA synthesis at the time of labelling, despite RdRp nsps also co-occurring in the lumen of DMVs [90]. Similar findings in studies with γ-CoVs showed that less than 1.5% of RdRp nsps colocalized with dsRNAs [94]. These results suggested that dsRNAs and minor localizations of RdRp nsps are not bona-fide markers for viral RNA synthesis. Rather, encapsulating dsRNAs within DMVs may be used to protect the replicase from activating immune pathways.

Another proposed structure for RNA synthesis is the DMS-ER network. DMSs have been detailed to also occur in α-virus Semliki Forest Virus (SFV) of the Togaviridae, and they are reported to be sites of RNA synthesis [113]. Due to the small openings in DMSs that connect them to the cytoplasm, it was hypothesized that DMSs would be probable sources of RNA synthesis in CoVs since they also occur in abundance in α-CoVs [94], δ-CoVs [PDCov], γ-CoVs [IBV], and recently also, in low quantities, in β-CoVs [92]. However, actual labelling of nascent RNAs with radioactive ^3^H-U and EM imaging revealed little to no localization mid-late infection near zippered ER/CMs or DMSs in IBV, MERS, and SARS [92]. Intriguingly, both ^3^H-U and indirect immunogold-BrU labelling still supported membrane structures in proximity to DMVs as the primary regions of RNA synthesis [114], and not DMSs or CMs [92]. However, these conclusions were drawn only from images taken in mid to late infection and do not represent the dynamic findings of previous CoV RNA synthesis. Although DMSs tend to occur abundantly in IBV, RNA synthesis and virion production were not hindered in the pathogenic M41 strain of IBV that produced significantly less DMSs [100]. This could mean that very few DMS structures are necessary for subsequent RNA synthesis and virion production. Since (1) DMSs occur in low abundance with non-γ-CoVs, (2) they are difficult to discern among other membrane rearrangements, and (3) their time of production remains in question, it is possible that very few DMSs are required in early stages of infection. As the infection progresses, late RNA synthesis could migrate from DMSs or CMs to DMV-associated structures. Since IBV DMVs have repeatedly been reported to occur as lone vesiculations with only a few connected to the ER, it may not be a requirement for them to be connected to the ER as previously stated. Still, RNA synthesis has also been reported to have low background activity in the cytosol of infected MHV cells [114], and ^3^H-U or immunogold-BrU labelling with EM may not be sensitive enough to indicate minor regions of RNA synthesis. Following similar reasoning, the existence of CMs that do not occur in some CoVs may perform functions similar to those of DMSs. Alternatively, CMs may simply be consequential constructs caused by an overaccumulation of nsps despite their close relationship to DMVs. To elucidate the nature of RO-RVN formations and RNA synthesis, it would be best if a future study combines labels for TM nsps, RdRp complexes (nsp7 + 8 and nsp12), nascent RNA labelling, dsRNA labelling and EM all together throughout several time points spanning early to late infection with high resolution imaging. Of course, all results and findings are at the mercy of sample preparation and handling to carefully preserve the delicate nature of these RVNs.

This unpairing between dsRNAs and RdRp complexes seems to be a common theme among all of the CoVs examined in this review. Even more curious is the observation that MHV nascent RNAs delocalize with dsRNAs between 4.15–5 hpi to 8–9 hpi, with a Pearson correlation coefficient dropping from ~0.6 to ~0.35 [62]. In the perspective of virion synthesis, statistical correlations between membrane rearrangements, CM-DMV-virion abundance in MHV and DMV-DMS-virion abundance were low [100]. However, these findings still do not exclude DMVs from being involved in RNA synthesis, as EM imaging revealed that nascent RNAs still localize near DMVs and the ER in early and late infections [90,92]. Determining how dsRNAs collect inside the lumen of DMVs has also been troubling since no imaging provided has been able to detail DMV intermediates so far.

Returning to the pore hypothesis, recent cryo-EM tomograms of DMVs revealed a pore complex formed by nsps, sparsely scattered on DMV surfaces, opening the lumen to the cytosol in MHV and SARS-CoV-2 infected cells [115]. The pore has 6-fold symmetry, and the channel begins with a 6 nm wide opening facing the cytosol, surrounded by a protruding crown with 6 prongs extending 13 nm outward and 14 nm away from the central axis of the opening. The pore is stated to be analogous to the reoviridae genome packaging pore [115]. RNA export function has yet to be confirmed, but the structure and its 6-fold symmetry is primarily composed of nsp3, which has RNA binding capacities [106]. The luminal side of the pore complex appeared denser, and Wolff et al. speculated that other luminal DMV-associated proteins such as N and/or nsp12 RdRp associate with this pore. Since nsp7, nsp8 and nsp12 form a reverse transcription tunnel, it is possible that the viral transcriptome associates with the luminal nsp3s of the pore. This suggestion is additionally supported by nsp3′s known function as a scaffold protein for other nsps in the RTC [116]. Such a complex could then transcribe RNAs and export them upon synthesis in an efficient manner. Despite the pore being composed of nsps with catalytic activity, the pore itself has no confirmed catalytic activity, characteristic of other viral portals, such as those in bacteriophages, Reoviridae and Herpes [117,118,119]. We thus suggest the RdRp could be used as a motor to feed transcripts into the pore. Meanwhile, aborted, or malfunctioned transcripts are not exported, but instead are left within the DMVs, due to the size limitations of the pore. Confirmation of this structure’s existence is a major step forward to completing the coronavirus RO puzzle. It is also yet to be confirmed if equivalent pores exist in CoVs outside of the β-CoVs.

It is possible that after significant suppression of host innate antiviral pathways due to the accumulation of nsps or other ORF proteins, that RNA replication no longer needs to reside in perinuclear membranes and may migrate throughout the cytosol during mid-late infections. This pore has been confirmed only in the β-CoVs, MHV and SARS-CoV-2, but due to the phenotypic similarity between many β- and γ-CoVs, it is likely that pores exist beyond the findings of Wolff et al. [115].

## 8. Viral Proteins–Structures, Expression and Assembly

### 8.1. The Nucleocapsid (N) Protein: Genome Packaging

The nucleocapsid (N) proteins of coronaviruses are reasonably well conserved proteins, although the SARS-CoV-1 and SARS-CoV-2 N-proteins, nearly 90% identical to each other, show only about 25% sequence identity with those from other members of the *Coronaviridae* family [120]. Nevertheless, most of them exhibit only moderate variation in size, usually being between just below 400 amino acyl residues (aas) (e.g., Porcine transmissible gastrointestinal CoV, TEGV, of 382 aas), to just over 450 aas (e.g., Murine CoV-3, MHV3, of 454 aas) [121]. They have three domains, an N-terminal domain (NTD) nearly 200 residues in size which is the dominant RNA-binding domain, a central Ser/Arg-rich flexible linker domain with a striated box of about 50 residues, and a C-terminal domain (CTD), which like the NTD, is roughly 200 residues in length, but functions in dimer/oligomer formation [122]. The N protein has primary functions in self dimerization/oligomerization and RNA binding, yet although the NTD serves a primary function in RNA binding, all three domains have affinity for nucleic acids [123]. In addition, there are intrinsically disordered regions near the N- and C-termini of these N-proteins, each about 50 residues in length [124].

N proteins have multiple functions including but not limited to: (1) forming stable but dynamic complexes with the genomic RNA for compaction of the nucleic acid in the viral particle, (2) interacting with the structural membrane (M) protein to promote membrane envelop folding and virion assembly, (3) interacting via two distinct regions of N with the non-structural protein, nsp3, to allow proper recruitment of N to the replication/transcription complex, (4) playing an essential role in enhancing the transcription of genomic RNA and viral mRNA, (5) increasing RNA replication efficiency, in part, by facilitating separation of the two RNA strands, (6) interfering with host cellular defense processes such as interferon production, and (7) promoting host cell death (apoptosis) [125,126,127,128]. The N-protein can be phosphorylated to facilitate condensation with RNA and the M-protein and to modulate the liquid-liquid phase separation [129,130]. As noted above, its recruitment to the RTC plays a role in the coronavirus life cycle [131].

The details of many of these functions have been elucidated to a considerable degree, and several of them are clearly interrelated [123]. Its recruitment to RTC plays a crucial role in the overall coronavirus infection cycle [131]. Self-association of the N-protein, which also depends on its RNA binding capacity [132], is required for formation of the viral capsid, which occurs at intracellular membranes of the ER-Golgi intermediate compartment. Unfortunately, many details of the molecular packaging inside the virion have not been fully elucidated. Early electron microscopy revealed that the ribonucleoproteins (RNPs) are helical, consisting of coils of 9–16 nm in diameter with a hollow interior of about 3–4 nm [121]. In the mature virus particle, the capsid protects the viral genome from caustic chemicals and extreme physical conditions [133]. In this regard, it is important to note that N has protective RNA folding/chaperone activity, due in part, to the central disordered domain (the LKR domain), reducing the free energy barrier for dissociation of the nascent minus RNA chain from the genomic RNA template during discontinuous RNA transcription. N also promotes template switching, which may be a primary cause of its acceleration of transcription [134].

The 3-D structure of N together with NMR analyses revealed that the basic region between aas 248 and 280 in the SARS-CoV-1 N protein binds RNA, while the region just C-terminal to this sequence promotes octamerization of the CTD [135,136,137,138]. The former region forms a positively charged groove, being able to accommodate either single stranded or double stranded negatively charged nucleic acids [139]. Such interactions allow formation of a compact ribonucleoprotein complex, the nucleocapsid, that ensures timely replication, reliable transmission and proper regulation of translation while in the cell, before formation of the filamentous nucleocapsid of about 12 nm in diameter and up to several hundred nm in length, that will be incorporated into the viral particle during assembly [123]. The assembly process also depends on the M protein, which together with the E protein, is a primary core constituent in the final virion. By using 3D cryo-electron tomography with MHV particles, it was possible to see that the viral membrane was nearly twice the thickness of a typical cell membrane, possibly due to the C-terminal domain of the M protein [140]. It should be clear that the ribonucleoprotein complex, together with the closely associated M-protein, plays a major role in envelope formation and viral budding within intracellular ER-Golgi complexes (see the section on the M-protein). For this reason, the assembly of the N-protein oligomer with its associated RNA has been considered to be an appropriate target of drug action [141] (see below).

The N-protein has proven to be a successful target for antiviral drugs and may be useful for the potential development of vaccines. This topic has been extensively reviewed recently [142], and only a couple of examples will be provided here. Cyclosporin A and its non-immunosuppressive derivatives are effective antiviral agents for coronaviruses and many other viruses. They normally bind to cellular cyclophilins, thus inactivating the cis-trans peptidyl-prolyl isomerase activities of the latter. Cyclosporin A (but not cyclosporin B) binds to and blocks the interaction between the N-proteins of various CoVs and cyclophilin to prevent viral RNA replication. Thus, cyclophilin inhibitors such as cyclosporin A block this protein-protein interaction, inhibit replication, and thus prevent infectivity [143]. Examining several cyclophilin inhibitors revealed that even non-immunosuppressive cyclosporin derivatives can block replication, showing that they could be effective antiviral agents with minimal side-affects [143]. Clearly these compounds might prove effective at blocking diseases such as Covid-19.

In another recent study, Lin et al. [138] examined the structure-based stabilization of N-protein-protein interactions for the purpose of designing antiviral drugs. This unique approach for the discovery of novel drugs was based on the high resolution 3-dimensional structure of the N-terminal domain of the MERS-CoV nucleocapsid protein (N-NTD). Non-native interacting interfaces of the dimeric N-protein surface proved to form a conserved hydrophobic cavity that could be used for targeted drug screening. The authors evaluated the complementary surface as a potential binding pocket for drugs and identified 5-benzyloxygramine as an ortho-steric stabilizer that exhibits both antiviral and N-NTD protein-stabilizing activities. X-ray analyses revealed that 5-benzyloxygramine stabilizes the N-NTD dimer through hydrophobic interactions between the protein and the compound. This causes abnormal oligomerization of the protein. Thus, novel approaches can be used to identify potential drugs that can be used to fight viral infections [138].

In case the antiviral approaches discussed above do not prove successful in combating the current or any future pandemic, there may be a need for novel antiviral approaches that can target emerging viruses, particularly when no effective vaccine or pharmaceutical is available, as is currently the case for Covid-19. Abbott et al. [144] showed that a CRISPR-Cas13-based strategy, which they called PAC-MAN (prophylactic antiviral CRISPR in human cells), can be used for viral inhibition by effectively degrading viral RNA in intact cells. The approach was tried against SARS-CoV-1 and live influenza A virus in human lung epithelial cells. CRISPR RNAs targeted conserved regions of the target proteins and proved to reduce viral load. The authors concluded that a set of only six CRISPR RNAs could target more than 90% of all coronaviruses, thus being potentially applicable to diseases caused by both human and animal coronaviruses. This technique could be developed for safe and effective delivery into the respiratory tracts of intact animals [144].

### 8.2. The Envelope (E) Protein: Viral Assembly

Among the essential conserved transmembrane proteins in the Coronaviridae family, the Envelope (E) proteins are multifunctional viroporins. The genomes of CoVs may encode up to 2 additional viroporins, 3a and 8a, making CoVs among the most viroporin-rich RNA viruses. E proteins are 74–109 aas long [145] with multiple domains and cellular associations. The N-terminal end is a short hydrophilic region followed by a hydrophobic region containing the α-helical trans-membrane-spanning segment (TMS). Following this TMS is a C-terminal hydrophilic region. E proteins contain an unusually short, palindromic transmembrane helical hairpin around a pseudo-center of symmetry, a structural feature which seems to be unique to CoVs [146]. The hairpin deforms lipid bilayers by way of increasing their curvature, providing a molecular explanation for E protein’s pivotal role in viral budding [147]. Depending on the CoV, E protein may be glycosylated [148], palmitoylated [149] and ubiquinated [150]. These conclusions have been extensively confirmed [151,152,153], although unfortunately, a high-resolution x-ray or cryoEM structure is not yet available. Deletions in various parts of the E protein throughout its length produces an attenuated virus.

Expression of the E and M proteins together in transfected cells is sufficient for VLP formation in MHV, TGEV, BCoV, IBV and SARS-CoV-1 [149]. In some CoVs, their expression is not essential to produce intracellular particles; however, their loss may result in a severe reduction of the number of released virions from the Golgi as for SARS, MERS and MHV. In many CoVs, deletion of or mutations within the E protein gene attenuates the virus both in vivo and in vitro and reduces the progression of disease and mortality in animal models. For SARS, ΔE mutants are able to replicate viable particles albeit at a lower efficiency [154]. HCoV-OC43 ΔE mutants have a dramatic deficiency in viral replication but are still able to replicate at a much lower efficiency than wildtype in CNS tissue culture and in mice with decreased pathogenicity [155]. TGEV ΔE and MERS ΔE mutants completely lose their ability to bud from host cells, making intracellular virions that are unable to infect new cells [156,157]. Deleting E from IBV results in lethality for the virus [24].

The ion channel activity of an E protein is a major contributor to the hallmark inflammatory response [158], leading to the cytokine storm and acute respiratory distress syndrome (ARDS) associated with respiratory CoV infections [159]. The two other recognized ion channels, 3a and 8a, have also been shown to illicit inflammatory stress in a similar manner; however, targeted changes to E have the strongest attenuation of viral infections in SARS [160] and MERS [157]. Some CoVs, such as γ-CoV IBV do not have other viroporins [24], and studies on IBV E have shed some light on the various responsibilities of E proteins during infection without the noise of other accessory viroporins.

E proteins are largely interchangeable between β- and γ-CoVs, but not between E proteins of these CoVs and the α-CoVs [161]. Structurally, β- and γ-CoV Es are more similar to each other than they are to α-CoVs E proteins, containing predicted β-hairpin structural motifs in their C-terminal cytoplasmic facing tails (see preceding paragraph), responsible for localizing the protein to the membrane. In all CoVs, E localizes to the ER/ERGIC/Golgi perinuclear membranes, consistent with the CoV-induced reticulovesicular network. Specifically, the cytoplasmic tail of IBV-E targets and binds to the golgi tag GM130 and trans golgi tag p115, while SARS E localizes with GM130, ERGIC tag ERGIC53 and trans-Golgi tag p230 [162,163]. Surprisingly, neither the TMD nor the β-hairpin motif is necessary for Golgi localization as shown with truncated SARS E, suggesting the presence of a second Golgi localization tag within the N-terminal region of E. A truncated SARS-E, containing its N-terminus attached to the C-terminus of the VSV G protein still localized with GM130, ERGIC53 and p230 [163].

All CoVs have a well conserved proline residue in the β-strands of E tails. Mutating Pro54 to alanine (P54A) in IBV E disrupts its localization with the Golgi [163]. While the E protein is produced in abundance during infection, very few copies are actually incorporated into mature virus particles [148,158]. Despite this fact, E deletion mutants of SARS, MERS, MHV, TGEV and IBV produced weakened viruses that could either not escape cells (MERS, IBV, TGEV) [24,156,157], or had difficulty budding (SARS, MHV) [154,164]. For SARS, deleting the E protein leads to 100–1000-fold lower viral titers in lungs and nasal turbinates of infected hamsters [154] and lower NFκB (a major immune transcription factor) activation [158]. SARS with both 3a and E proteins deleted were non-viable but were rescuable if either 3a or E was reintroduced [160], thus suggesting marginal flexibility and exchangeability in viroporin roles.

Like many of the other CoV proteins, E may perform multiple roles during the infection process. Since it is only incorporated into virions in small numbers, it has been proposed that E helps scaffold newly forming virions, adding to their structural integrity. EM scans of intracellular virions revealed no change in SARS morphology upon deletion of the E gene, but upon viral purification, many of the ΔE mutant viruses had aberrant or misshaped morphologies, suggesting that an E deficiency makes CoV particles susceptible to shearing forces [154]. In all E deleted CoVs, smaller in vitro plaques are observed, and viral titers are reduced [148].

When purifying IBV E protein, two distinct molecular weight pools for the protein were extracted, suggesting oligomerization properties [165]. The lower molecular weight pool was predicted to consist of monomers and/or homodimers, while the higher molecular weight pool was predicted to consist of homopentamers, or possibly, hetero-oligomers associated with host proteins [165]. Wild type IBV E tended to favor the higher molecular weight pool, suggesting that a majority of the resulting conformations were homopentameric. Disrupting the hydrophobic domain composing the TMD of IBV E produced no high molecular weight pool, implying that the pentamers are formed through α-helical interactions [165]. Homopenatmers of E had already been suggested from earlier studies [166]. Indeed, it was predicted that the E protein requires homopentamerization for ion channel (IC) activity, where the amphipathic/hydrophobic α-helical TMSs form a continuous channel just large enough (4–5 A for SARS) to fit a dehydrated cation (H^+^, Na^+^, K^+^_,_ or Ca^2+^) through it [167,168].

β-CoV Es tend to be selective for Na^+^ over K^+^ ions, but conflicting evidence suggests that SARS E is slightly selective for K^+^ and Ca^2+^ ions and is dependent on the slight negative charge of ER membranes [168,169]. However, because the ER and Golgi are large Ca^2+^ stores in cells, it is more probable that E functions as a Ca^2+^ efflux channel, while minimal amounts of Na^+^ and K^+^ are imported into the ER/Golgi lumen [168]. For SARS, the predicted residues responsible for conferring IC activity to E are N15 and V25 [167]. Solution NMR analyses in dodecyl-phosphatidylcholine micelles revealed dynamic conformational changes in the homopentamer that could accompany cation translocation [164].

The SARS E N15 residue and its polar equivalents in other CoVs may provide a cation selectivity filter, while V25 and V28 form a 2.0–2.3 Å constriction, predicted to correspond to the closed state of the IC [167]. In fact, creating a recombinant SARS E virus with a mutated residue at position 15 (N15A), but not at position 25, consistently eliminated IC activity and reduced pathogenicity in mice [170]. V25F mutants reverted back to the IC^+^ phenotype either by directly mutating F25 to C, or by mutating neighboring residues: L19A, F20L, F26L, L27S, T30I, and L37R 2 dpi in mice. Modifying the equivalent IC residues (T16A or A26F) in IBV E in vitro gave similar results, with E-A26F unable to form VLPs in vitro. However, no reversion mutations were recorded [24]. The reversion of V25F to various other residues just after 2 dpi in mice revealed an obvious danger in generating attenuated point mutant viral vaccines. However, analysis of the N15A mutant, which did not mutate back within the time interval of the study, suggests that mutations in the predicted cation selectivity filter are more lethal and specific than the structural V25/V28 residues. Perhaps leaving the filter intact causes the pore to retain selectivity, and opening the channel, obstructed by V25F, is easier than reverting the N15A filter residue.

The drug, hexamethylene amiloride (HMA), shown to abolish IC activity in MHV E and HCoV-299 E, also inhibits the IC activity of SARS E, likely by associating with the N15 residue and the equivalent residues in other CoVs [167]. Further investigations into SARS N15A mutants revealed 80–100% survivability rates in mice, despite similar disease progression during the first 2 dpi [170]. Overall, lung autopsies of infected mice revealed less swollen alveolar walls and airways free of pulmonary edema, as opposed to the typical Acute Respiratory Distress Syndrome (ARDS) phenotype induced by SARS [170]. Furthermore, neutrophil recruitment was lower in N15A mice due to the reduced amount of secreted IL-1β, TNF and IL-6 proinflammatory cytokines. IC activity also promotes the fitness and release of IBV viral particles [171,172]. HMA treated MHV or HCoV-299 infected cell lines exhibited much smaller plaques as opposed to the HMA-free infected cells with plaques roughly 3–4 mm in diameter [173].

In addition to inducing pro-inflammatory responses, the IC activity of E may confer major modifications to secretory or apoptotic pathways. IBV infections are associated with p53-independent, caspase-dependent, CHOP transcription factor and IRE sensor-mediated unfolded protein response (UPR) pathway-regulated apoptosis. This pathway is stimulated by ER stress, marked by the cleavage of downstream poly ADP-ribose polymerase (PARP) [24]. Similarly, SARS induces apoptosis in cell cultures via protein kinase R (PKR) [174], caspase-3-mediated ER stress, JNK-dependent pathways [175], and PERK and eIF2α-mediated UPR activation [176]. However, CoVs prefer nonapoptotic budding of virions and regulate the apoptotic pathways, likely through E protein IC activity to optimize virus release. Specifically, CoV-induced apoptosis is related to ER stress, induced by the viral replication in the ER-derived RVN and the expression of unfolded, unprocessed accessory and structural proteins [24,177]. In SARS, S induced the greatest ER stress [176], although the E IC activity may also induce stress to a lesser degree late in infection [24,178].

E IC deficiency in CoVs, induced by mutations or drugs, leads to smaller plaques in in vitro tissue cultures and reduced pathogenicity in vivo. When eliminating E IC activity by generating recombinant IBV E-T16A or E-A26F mutants, levels of cleaved PARP and pro-inflammatory mRNAs for IL-6 and IL-8 were reduced [24]. SARS E protein alone reduced ER stress of vero-E6 and MA-104 cells when the stress was induced externally by adding either respiratory syncytial virus or an ER stress inducing drug, tunicamycin or thapsigargin. In comparison to wildtype, SARS-CoV-ΔE underwent higher rates of apoptosis and increased the expression of double specificity phosphatases, DUSP-1 and DUSP-10, despite expressing lower levels of proinflammatory chemokines CXCL2 and CCL2 [179]. DUSP-1 and DUSP-10 negatively regulate mitogen-activated protein kinase (MAPK) signaling, reduced the viral induced inflammatory response, and reduced the synthesis and secretion of TNF, IL-6, CCL2/MCP-1, CCL3, CCL4 and CXCL2/MIP-2 [179]. Thus, deleting E in SARS or IBV leads to a weakened proinflammatory response while also attenuating virus production and infectivity. Possibly, E allows host tolerance to the virus during early-mid infection, preventing apoptosis for a long enough period to allow production of more viral particles through budding.

Meanwhile, wild type SARS had reduced expression of ER stress induced GRP78, GRP94 and MHCI antigen-presenting facilitator HSPs on the surfaces of infected cells compared to the mutant [179], and as noted above, SARS-CoV-ΔE deletion mutants induce apoptosis in infected cells at a greater rate than their wild type counterpart [179]. By contrast, late infection IBV IC activity may induce apoptosis by destabilizing the ion gradients between the Golgi lumen and the cytosol [24]. PEDV E protein was also reported to induce ER stress through the UPR, but the results were attained through a transfected plasmid encoding only PEDV E [178]. The IC activity of E protein [168] as well as those of the other accessory viroporins, 3a and 8a [160], activate the NLRP3 inflammasome by effluxing Ca^2+^ from the lumen of the ER/ERGIC/Golgi, altering the homeostatic levels of cytosolic Ca^2+^ [168,180] and resulting in upregulation and secretion of pro-inflammatory TNF-α, IL-1β, IL-6 and IL-18 [160,181]. ER stress [182] and ROS production [183] are also activators of the NLRP3 inflammasome, and due to E protein’s regulation of ER stress, they may also activate NLRP3 through an alternative mechanism.

Release of assembled CoV virions requires secretory pathways during early-mid infection, and lysosomal pathways for egress in late infection [62,182,184]. While secretory pathways are necessary for production of CoV structural proteins and processing, the Arl8b-dependent lysosome exocytosis pathway has recently been shown to be the exit pathway of mature MHV and SARS-CoV-2 virions [184]. Hence, E protein’s role in viral release may be most important during assembly, before egress of mature virions to the cellular membrane. It is possible that E protein modifies secretory pathways to facilitate the release of intracellular particles since E mutants of several CoVs have difficulty leaving the cell. IBV infected cells, or expression of E alone, induces the neutralization of the Golgi pH, suggesting a role for E in altering secretion [185]. Replacing the hydrophobic domain in IBV E with VSV Glycoprotein HD led to a decrease in viral shedding, increase in damaged particles and accumulation of prematurely cleaved S protein [185], all suggesting a protective role for E in the maturation of other structural proteins. On the other hand, merely replacing the residues in the HD responsible for IC activity in IBV did not affect glycosylation or proteolytic processing of S [24]. Hence, the HD domain in its entirety, but not IC activity of E alone, may contribute to some of the purported functions of E. This suggestion is further supported by the fact that monomers were more strongly correlated with IBV-induced secretory modifications than pentamers although IC activity supports virion assembly [165]. Thus, additional conformations of E may equip CoVs with multifunctional molecular tools.

E proteins also contain a C-terminal class-II hydrophobic PDZ binding motif (PBM) [158] that anchors them to lipid membranes and participates in the relocalization of syntenin-1, a multifunctional adaptor protein that is involved in trafficking of membrane proteins to perinuclear regions [186], thus, activating p38 MAP kinase-mediated inflammation [160]. This PBM has also been found to interact with the host PALS1 protein, an epithelial cell polarization protein [187], disrupting the tight junctions between epithelial cells. It may perform a role in non-apoptotic virus release through a cell to cell exit mechanism [188]. Deletion of the PBM in SARS, either by truncating the E protein at the C-terminus or by replacing the residues within the PBM to produce a mutant E of the same length, did not affect viral replication efficiency in vero-E6 and DBT-mACE2 cells [189]. However, mice infected with virus possessing E, but lacking the PBM, did show a decrease in expression of inflammatory cytokines and active p38 MAPK in their lungs, reducing the pathogenic response and mortality [160,189]. Additionally, SARS transfected vero-E6 cell lines in which the full length PBM of E was disrupted, and deleting 3a (Δ3a, E-PBM^-^), or its inverse (3a-PBM^-^, ΔE) resulted in an infectious virus [160]. Introducing a stop codon to truncate the E protein missing the PBM (3a, E-ΔPBM) reverted back to wildtype [160], showing an ability of different CoV PBMs to substitute for each other. These studies revealed the essentiality of PBMs in SARS-CoV-1 infections. HCoV-OC43 E PBM greatly improves propagation in human and mouse neuronal cells and infectivity in the brain and spinal cords of mice, and its removal attenuates the virus [155].

Other mutations introduced into the E protein also promoted attenuation of the virus, implying that other important regions of the E protein contribute to viral pathogenicity. Deleting regions along the hydrophilic C-terminus of SARS E led to reduced pathogenicity, although deletions at the very end of the C-terminal tail had no such effect in a mouse-adapted model [190]. Clearance of viral infections, typical of CoV survivors, is associated with elevated levels of T cell production [191]. Infection by SARS is in part attributed to reduced numbers of T cells, primarily CD4+ T cells, leading to the host’s inability to clear the infection [190,191]. Attenuating the SARS E protein by deleting these regions leads to less lung tissue damage and higher T cell counts, likely disrupting the β-hairpin Golgi localization motif or the PBM [163]. In MHV, replacing clustered charged residues within the C-terminal end of the E protein with alanine (E-K63A/K67A or E-D60A/R61A) resulted in thermally unstable virus particles with much smaller plaque morphologies [192].

### 8.3. The Spike (S) Protein: The Primary Receptor and Membrane Fusion Mediator

Spike (S) proteins of coronaviruses are the receptor binding glycoproteins and class I fusion proteins of CoVs. S proteins are large [1162–1376 aas], are synthesized in the ER/ERGIC, and are post translationally modified in the Golgi, undergoing proteolysis and extensive O- and N-linked glycosylation as well as palmitoylation. SARS S proteins are relatively unique among CoVs, sharing little sequence similarity with their relatives despite strongly conserved structures and functions [193]. SARS-CoV-2 S is similar to its ‘predecessor’, SARS-CoV-1 S, with a 76% aa identity with SARS-Urbani S and 80% identity with bat SARS-CoV ZXC21 S and ZC45 S [194], and 98% identity with bat RaTG13 [32,195], conserving several N-linked glycosylation sites [194]. During synthesis, the protein may be cleaved into the S1 (head and receptor binding) and S2 (membrane embedded stalk and fusion) subunits by either host or viral proteases [196], or it can be left as a full-length S protein, requiring cleavage at S1/S2 upon receptor binding [197]. If cleaved, these subunits then remain noncovalently bound to each other [193]. The S1 subunit can be further divided into the N-terminal domain (NTD) and C terminal domain (CTD), both of which participate in receptor binding [198]. The S1/S2 structure then trimerizes with 2 other S1/S2 molecules to form the complete S protein [196,198].

There are multiple important domains among the S1 and S2 subunits that contribute to the binding and fusion functionalities of the protein. The S1 subunit (N-terminal residues 14–685) for SARS-CoV-2) [199] contains the receptor binding domain (RBD) that associates with the host receptor (DPP4 for MERS or hACE2 for SARS). Comprising the RBD is either the S^A^ domain of CoV-HKU1, HCoV-OC43 and MERS or the S^B^ domain of SARS-CoV-1 and SARS-CoV-2, which interacts with the host receptors [194]. S^B^ directly binds to ACE2 to allow viral entry of target cells [194]. The S^B^ domain operates like a lock and key, existing in an open or closed conformation which possibly induces differential folding at the S1/S2 junction [194,200]. The open and closed conformations of S^B^ are transient states, stochastically revealing and sheltering the RBD [195]. For SARS-CoV-2, the closed conformation is indicated by the RBD bound in trans in a pocket provided by the NTD and RBD of the neighboring S1 monomer [201]. SARS-CoV-2 S has been reported to have an even higher binding affinity [202], about 20-fold higher than SARS-CoV-1 S, for the ACE2 receptor [195]. SARS-CoV-1 S also binds to the C-type lectin DC-SIGN (dendritic cell specific intercellular adhesion molecule grabbing nonintegrin) as well as DC-SIGNR of dendritic cells without engaging the fusion complex [203]. Since dendritic cells migrate to lymphatic tissues, SARS may utilize dendritic cells as ‘ferries’, traversing blood and lymphatic vessels to new ACE2^+^ tissues, leading to systemic infections [203].

Upon binding to ACE2, the S^B^ domain goes into the open configuration, releasing constraints at the S1/S2 site [194]. SARS-CoV-2 is much more susceptible to fusion activation than is SARS-CoV-1, indicating the presence of an additional furin cleavage site, confirmed to exist between S1 and S2 (residues 677–687) [194]. Depending on the mode of entry and the CoV strain, the furin cleavage site can be cleaved by host furin, transmembrane protease serine protease-2 (TMPRSS-2), TMPRSS-4, trypsin, lysosomal cathepsins or airway trypsin like protease (HAT), priming the class I fusion complex during synthesis, either at the cell surface or within an endosome [32]. The additional furin cleavage site in SARS-CoV-2 S may expand its tropism or propensity to fuse with host cells. Many SARS-CoV-1 or SARS-CoV-2 pseudo-virions/virions contain pre-cleaved S1/S2, indicating that cleavage can occur during S synthesis [197,199].

Fusion is likely pH-independent [204] but may be regulated by endosomal maturation and Ca^2+^ [51,52,205]. After binding to the receptor and cleavage of S1/S2, a final cleavage must occur at the S2′ site (residue R797 in SARS-CoV-1) [206]. Once cleaved, steric bulk is released from two amphipathic α-helical, 4-3 coiled-coil heptad repeats, HR1 (residues 910–988 in SARS-CoV-2) and HR2 (1162–1206 in SARS-CoV-2), in the S2 subunit, releasing stiffness from this joint. X-ray crystallography revealed that the post fusion conformation of these HRs is characterized by a hip-knee-ankle style folding, where three HR2 helices collapse onto the hydrophobic grooves in an antiparallel manner of the central coiled-coil of the HR1 helices [194,207]. This folding reduces the distance between the viral envelope and the host surface/endosomal membrane, allowing insertion of a fusion peptides (FP) into the host membrane before membrane fusion [206].

Several efforts have been made to identify the FPs and regions in S2 that contribute to membrane fusion. Three regions in SARS S2, termed R1, R2 and R3, were found to have membrane-associating properties. R1 (858–886) is upstream of HR1, R2 (1077–1092) is situated between HR1 and HR2, and R3 (1190–1202) is proximal to the TM portion of S2. Of these regions, mutations in R1 led to a decrease in syncytia formation [206]. Upstream and overlapping with R1 are regions discovered through Wimley and White interfacial hydrophobicity analysis. They are called WW-I (770–778) and WW-II (864–886) and are strongly associated with membranes [206]. An exposed FP (FP1 798–818) is likely to occur in the WW-I N-terminal side of HR1, containing Ca^2+^ salt bridge-forming residues, D830 and L831. Immediately following FP1 is FP2 (816–835) which contains two disulfide bridge-forming cysteine residues C822 and C833. FP1 is highly conserved among CoVs with a crucial invariant LLF motif that when mutated to alanines causes defective fusion [208]. Additional putative FPs are the Alt-FP (770–788), overlapping with the WW-I, downstream of the S1/S2 cleavage site and internal FP (IFP) (873–888), coinciding with R1 [209]. Additionally, the region upstream of the TMS (1185–1202) seems to have membrane association properties [209]. ESR analyses conducted on each of these lone peptide segments, exposing multilamellar vesicles (MLVs) to these putative FP segments, induced an increase of membrane ordering, an indicator of viral fusion peptide activity. In all FPs, a requirement for Ca^2+^ was observed, and activity occurred from pH 5 to pH 7, consistent with the ability of S to fuse membranes at neutral pH values. FP1 and FP2 likely work in concert with each other, embedding themselves into membranes, forming disulfide and Ca^2+^ salt bridges to stabilize the fusion complex [206,209]. Fusion activity of FP2 was undetectable when Ca^2+^ was not present or if the complex was treated with disulfide bond reducing dithiothreitol [209]. Understanding their conformations may provide additional drug targets for the inhibition of SARS entry.

S protein also induces cell to cell fusion with production of syncytia with tissue damage [210]. New intracellular CoV particles can exit a cell and enter directly into adjacent cells of epithelial tissues, leading to disruption of cellular barriers and production of multinucleated cells. SARS-CoV-1, MHV, IBV, MERS and SARS-CoV-2 have all been reported to induce formation of syncytia in vitro in cell lines, likely due to S protein’s ability to engage its fusion complex at neutral pH [32,211,212,213,214]. However, SARS-CoV-2 has an unprecedented capacity to form syncytia, producing multinucleated cells with hundreds of nuclei per 293T or Huh-7 cell [199]. Host proteases on membrane surfaces may be required for immediate fusion and entry of the virus into neighboring cells. TMPRSS-2, present on the opposite side of the membrane as S, seems to be required for syncytium formation in SARS-CoV-1 and SARS-CoV-2-infected veroE6 cells [214,215]. Lone SARS S, expressed via a cDNA in one of several cell lines, expressed S on the surface of transfected cells, and for some CoVs, syncytia have been reported to be formed by S alone [212]. These results suggest that SARS S binds to ACE2 and is readily cleaved by membrane bound proteases [215]. An accumulation of secreted S protein on the surface of cells may induce fusion between neighboring cell membranes.

Further supporting the requirement of a membrane protease to generate syncytia is work conducted with MHV-2, a strain of MHV with S that can only be cleaved by cathepsins and cannot generate syncytia [211]. TMPRSS-2 was identified as a potent entry factor for SARS-CoV-2 in nasal epithelial cells [216], and TMPRSS-2 is abundantly distributed in respiratory epithelia [217]. Hence, developmental therapeutics, suppressing the interaction of SARS with TMPRSS-2, may reduce viral replication in tissue. Additionally, mutating all 9 palmitoylated residues in S, which does not disrupt folding, trafficking or core functions, does disrupt syncytium formation [218]. The increased capacity of SARS-CoV-2 to form syncytia could be due to the additional furin cleavage site and the increased occurrence of pre-primed S protein, so that only the S2′ cleavage site is a prerequisite for fusion. Recently, it was discovered that pan-coronavirus fusion inhibitor, EK1 peptide variant EK1C4, could inhibit SARS-CoV-2 fusion in a dose-dependent manner by binding to HR1, but the exact mechanism was not revealed [199].

Tropism for ACE2 may be mediated by cholesterol. SARS and other enveloped viruses such as HIV have been reported to be dependent on lipid rafts for entry [219]. ACE2 associates with detergent-resistant cholesterol-rich microdomains in membranes, but treating cells with methyl-β-cyclodextrins (MβCDs) did not affect expression of ACE2 [220]. Cholesterol depletion in several cell lines treated with MβCD inhibited binding of S to ACE2 and SARS entry [220,221]. ACE2 receptor binding may be promoted by S palmitoylation, since palmitoylation was found to promote S association with lipid rafts and detergent-resistant membranes [218]. However, due to the virus’ ability to infect new cells through cell-to-cell fusion, depletion of cholesterol and lipid rafts could not completely suppress viral replication [219]. Thus, lipid rafts are promoters of viral entry, but not necessarily of viral replication within tissues.

It was noted earlier that S can bind to DC-SIGN, a C-type lectin on the surfaces of dendritic cells. However, SARS has also been known to infect monocytes through both ACE2-independent and ACE2-dependent mechanisms in the lungs of SARS patients [222,223]. Infection of T cells by SARS-CoV-2 has also been reported [224]. This may contribute to the severe inflammation and depletion of T cells. A preprint study detailed tropism for white blood cells which may be mediated through CD147. However, further evidence is required for confirmation [225]. While abundant IFN-γ, present in macrophages, may suppress viral replication [222], SARS has multiple IFN-suppressing strategies that allow it to evade and silence innate antiviral activity in monocytes [223]. For instance, the virus can avoid detection from intracellular pattern recognition receptors (PRRs) such as MDA5 and RIG-I. These PRRs, which illicit a specific antiviral response upon detection of a virus, are either never activated or silenced by suppression of IRF-3 [223]. Rather than an antiviral response, cytokine secretions may be dominated by nonspecific inflammatory mediators that contribute to SARS-CoV-1 and SARS-CoV-2-associated diseases [223].

S protein may activate complement in early infections, which could explain the early onslaught of cytokines circulating in SARS patients. IgM, IgG, mannose binding lectin (MBL) or an alternative pathway may allow recognition of and binding to S, thereby activating complement. Subsequent activation of complement downstream pathways results in a flush of proinflammatory cytokines, possibly leading to a cytokine storm. Since complement can be activated directly by the presence of antigen, S protruding from viral particles may be a major contributor in the development of disease. Since S is abundantly glycosylated with N-linked mannosyl oligosaccharides, sequestering of SARS could occur early by binding MBL to S. MBL has been shown to bind to S in SARS-CoV-1 bearing pseudo-viruses, specifically at an N-linked oligomannosyl glycosylation site in the RBD. This critical localization within the RBD prevents S from binding to DC-SIGN, but not to ACE2 [226]. MBL was also found to bind to SARS-CoV-1 infected FRhK-4 cells and immobilize actual SARS-CoV-1 particles, inhibiting their infectivity [227]. Thus, low serum levels of MBL may be a susceptibility factor for the acquisition of SARS [227]. Despite these findings, it remains unclear what role MBL and C3b may have in activating complement early in the infection, even though complement activation has been confirmed in SARS-CoV-1, MERS and SARS-CoV-2 [228,229].

### 8.4. The Membrane Matrix (M) Protein, the Virion Scaffold

The homologous M proteins of CoVs and many other envelope viruses have been called the membrane proteins, the matrix proteins, the M proteins, or simply “M”. M is the most abundant protein in any one coronavirus virion, and it is among the most conserved and constrained of all the viral structural proteins [230]. This may be attributed to its many functions in the viral infection cycle as well as in interferon antagonism (see Table 3) [230]. All of the major structural proteins of these viruses are derivatized and/or hydrolyzed at specific positions by post translational modification (PTM) reactions (see Fung & Liu, 2018 [231] for a review). These derivatization reactions involve (1) protease-mediated hydrolysis by both virus- and host cell-encoded proteases, (2) either O (serine or threonine)- or N (asparagine)-glycosylation, and often both, (3) palmitoylation of the spike (S) and envelope (E) proteins, (4) protein phosphorylation by ATP-dependent protein kinases, and (5) ADP-ribosylation of the nucleocapsid (N)-protein. Other PTMs of nonstructural “accessory” proteins have also been documented [231].

The most complicated of these PTM reactions is glycosylation. M proteins of SARS-CoVs are O-glycosylated (on seryl and/or threonyl residues), and not N-glycosylated (on asparaginyl residues) [232,233]. The structures of the O-linkages are known and include O-linked N-acetylgalactosamine to which galactosyl and sialyl residues are glycosidically linked [234]. O-Glycosylation occurs in the Golgi and has been used as a marker for proper M protein intracellular trafficking, membrane insertion and maturation [235]. It seems that glycosylation is non-essential for assembly of some CoV virions, but it greatly facilitates the formation of active virus particles, and it also regulates interferon production (see below). The established or probable functions of M proteins are presented in Table 3.

The primary function of M is assembly of newly formed viral particles. As noted above, it is the most prevalent protein component of the virion. It provides a homo-dimeric scaffold for virion assembly and has affinity not only for itself, but also for all of the constituent structural proteins found in the virion. Thus, to provide the “master assembly function”, it has both homotypic and heterotypic associative properties. In one study, based on cryoEM, tomography and statistical analyses, Neuman et al., 2011 [236] suggested that M can assume two distinct conformations. One, they suggested, is elongated, being associated with rigidity, spike clustering and a narrow range of membrane curvature, while the other is more compact and is associated with greater flexibility and a lower spike density. Presumably, the proper ratio of these two forms determines the final virion construction. As noted above, M associates with itself to form dimers, but also with the nucleocapsid (N)-protein, the spike (S)-protein and the envelope (E)-protein as well as the genomic RNA. Thus, with M as the ‘glue’, holding the complex together, these primary constituents of the viral particle determine virion size and shape with M playing the dominant role.

A subsequent study led to the suggestion that initial self-assembly and ultimate release of the membrane-enveloped vesicle/particle (the virion) depends most importantly on the association of M with N and the viral RNA [237]. Assembly seems to be a multi-step process as illustrated in Figure 2: First, M self-associates, creating an M-protein homodimer, and this self-association process involves several distinct regions of M, explaining why this occurs with high affinity. Its heterotrophic interactions, then, may be largely responsible for the order of the protein associations. Because of the TMSs in M, this early intermediate is likely to already be membrane associated. Second, although M is made in the ER, it acts either in the trans-Golgi network, or the ER-Golgi intermediate compartment (ERGIC) during assembly, depending on the specific coronavirus under study, clearly requiring specific host-catalyzed trafficking of M through the endo-membrane network [238]. In this regard, it is important to note that two motifs in the C-terminal domain (DxEER and KxGxYR in the MERS CoV M protein) are ER export and trans-Golgi network retention sequences, respectively [238]. Third, M associates with N, the nucleocapsid protein, again probably via multiple sites in M, although a particularly important sequence for this association is the di-leucine motif in the C-terminal tail of the protein [237]. Based on mutational analyses, the N-terminal exo-domain or the central TMSs appear to be of lesser importance for the association of N with M. However, a central cysteyl residue (C158) also plays a role [237]. The C-terminal domain of N is largely responsible for the association with M [239]. Fourth, the M-N association allows the genomic RNA to become part of the developing particle complex because of the high affinity of N for this nucleic acid. However, the inclusion of the genomic RNA within the complex may occur simultaneously with step 4 because of the high affinity of N for the genomic RNA. Fifth, the endo-membrane-M association allows recruitment of the Spike (S)-protein to the particle. In fact, M has affinity for ALL of the other structural proteins that end up in the virion. Sixth, several M-protein residues seem to be involved in the final secretion and budding processes, and these residues are scattered throughout the protein, probably playing specific roles [237]. Finally, the E-protein, together with M, with which it interacts, plays a significant but less well-defined role in the assembly process [239]. The stage(s) of its involvement in the temporal scheme outlined here are not as well defined as the general scheme itself. As will become apparent, the viroporin functions of E and 3a are assumed to play a role (see below).

In addition to the associations with its own viral proteins, Gordon et al., 2020 [240] cloned, tagged and expressed 26 of the 29 SARS-CoV-2 proteins in human cells and identified the human proteins physically associated with each of the 26 viral proteins using affinity-purification/mass spectrometry (AP-MS). They identified 332 high-confidence SARS-CoV-2-human protein-protein interactions (PPIs). Among these, were 66 human potential drug targets (host factors), and these were targeted by 69 compounds. This work therefore provides a guide for the development of anti-viral drugs that may act against SARS-CoV-2 to block different aspects of the viral infection cycle.

## 9. Viroporin Activities: The E, 3a and 4a Proteins, and the Ability of Mutated M to Substitute for E

Three distantly related proteins in SARS-CoV-1 and other related coronaviruses display very similar hydropathy plots. The first of these is M (TC# 1.A.117), the second is protein 4a (TC# 1.A.89), and the third is protein 3a (which also can assume other designations, depending on the virus) (TC# 1.A.57). The similarities of their topologies can be viewed in Figure 3 as hydropathy plots. As shown in this figure, variations within each family occur, but they are similar in all three families (Figure 3). Moreover, surprisingly similar hydropathy plots with sequence similarity of borderline significance can be observed. As noted above, viroporin activities have been demonstrated for the E, 4a and 3a proteins as well as a fourth family of apparent viroporins classified under TC family # 1.C.99; however, porin activity has not been demonstrated for M [80,241,242]. Nevertheless, a most interesting study, suggesting a functional relationship between the M and E proteins, was conducted by Kuo and Masters [243]. E was eliminated by deletion of its structural gene in the mouse hepatitis virus. The virus was found to still be infective, but it showed poor assembly with altered virion morphology, and it gave rise to tiny plaques. The authors then selected for “suppressor” mutations with at least partially restored viral growth and virion production, giving rise to much larger plaques. The secondary mutations were found to be in M, and these mutants arose in a sequential process involving M-gene duplication, where one copy retained the native M gene while the second M-gene encoded an altered M protein (M*) with a truncated C-terminus. Both M and M* were incorporated into the virion. It seems that M* served as a surrogate for E, providing a new gene function through recombination. Since E is known to have viroporin activity [244,245], it seems plausible that M* had recovered (at least partly) the viroporin activity of the deleted E protein. Although these authors had a different interpretation of their observations, we suggest that the N-terminal transmembrane domain of M may be capable amino acid substitution that allows it to form transmembrane pores, a suggestion that needs to be confirmed or refuted. In Figure 4, we provide comparative hydropathy plots between the SARS 3a viroporin, HCoV 299E 4a viroporin and M protein.

## 10. Post Translational Modifications (PTMs) to Coronaviral Structural Proteins

### 10.1. Glycosylation

Glycosylation of viral proteins is a common theme among enveloped viruses due to their hydrophobic natures [251]. Glycosylation facilitates proper folding of the nascent polypeptide through recruitment of chaperone proteins and can play a role in cellular trafficking [251]. Cellular glycosylation also plays important roles in homeostasis and receptor signaling [252], which can be hijacked by viruses [251]. Viruses may utilize glycosylation to mimic or complement host proteins for receptor binding and entry, viral assembly/release, and/or immune evasion [251]. Several CoV E proteins and all S proteins contain N-linked and O-linked glycosylation sites. N-linked glycosylation is characterized by covalent en block binding of an N-linked 14-unit glycan precursor [253] onto an asparagine residue located within a recognition sequence Asn-X-Ser/Thr [254], where X is any amino acid except proline [251]. Attachment of the N-linked glycan to the Asn-X-Ser/Thr is performed by the enzyme, oligosaccharyltransferase, followed by further modifications by glycosidases and glycosyltransferases [231]. O-linked glycosylation involves the attachment of an oligosaccharide to the side chain oxygen atom of a serine or threonine residue, initially through the activity of N-acetylgalactosamine (GalNAc) transferase, and this reaction does not require a consensus sequence as does N-linked glycosylation. Rather than occurring en block, simple sugar residues are sequentially added after the initial O-linked addition. N- and O-linked glycosylations can occur simultaneously and do not compete with each other.

### 10.2. Palmitoylation

Palmitoylation is a common post-translational modification that occurs in the Golgi where palmitoyl transferases add a fatty acid, palmitate, to a cysteine residue via a thioester linkage [255]. Palmitoylation enhances the hydrophobicity of proteins and plays an important role in subcellular trafficking of proteins between membrane compartments [256]. Proteins can be readily palmitoylated, and the modification is typically reversible. Addition and removal of palmitate can play roles in protein and membrane regulation [256]. Palmitoylated proteins can be modified by either a single palmitoyl group, or dually modified with one or more palmitoyl groups and one or more prenyl or myristoyl groups. Palmitoylated proteins can be divided into four types, (1) single palmitoyl modifications, often at the end of a protein, (2) palmitoylation near a transmembrane domain, (3) dual palmitoylation and prenylation and (4) dual palmitoylation and myristylation [255]. Viral proteins may arrange on membranes in accordance to their palmitoylation status, and it affects assembly and localization of viral oligomers [257].

### 10.3. Ubiquitination

Ubiquitination is the process of adding the peptide, ubiquitin, to a lysine, serine or threonine residue in Ub or Ubl domains of a protein. Proteins can be mono or poly ubiquitinated, and the ubiquitination process is reversible through the action of deubiquitinating enzymes (DUBs). The process is performed by a sequential cascade of ubiquitin-activating enzymes (E1s), ubiquitin-conjugating enzymes (E2s) and ubiquitin ligases (E3s). Ubiquitination can regulate endocytic trafficking, influence inflammation, and target proteins to the proteasome for degradation. Viruses can take advantage of ubiquitination, often to avoid host cell defense mechanisms such as apoptosis, the type I IFN response, and MHC I antigen presentation. Many viruses including CoVs also have deubiquitinating proteins (SARS nsp3), although the biochemical functions of these proteins are not well defined [258].

### 10.4. A Focus on S Protein PTMs

As noted above, glycosylation of receptor binding viral glycoproteins is common among enveloped viruses. Spike (S) of Coronaviruses, Hemagglutinin (HA)/Neuraminidase (NA) of influenza, and glycoprotein 120 (gp120) of HIV are all receptor binding glycosylated proteins that protrude from the viral envelope to allow association with their corresponding cellular receptors. When S is expressed in the ERGIC, it is likely transferred to the Golgi and is co-translationally N- and O-glycosylated, and it then trimerizes if properly folded. Glycosylation plays a key role in successful folding, trimerization and secretion of the protein.

Due to their sizes, geometries, and extracellular natures, receptor binding glycoproteins are crucial targets in adaptive immune responses. Typically, humoral immunity, provided by serum antibodies secreted by B cells, sequester the released virus, preventing it from infecting new cells by binding to the viral glycoproteins or marking them for ingestion by patrolling phagocytes. Antibodies can be extremely specific and mimic the shape of the viral receptors, attacking the glycoprotein receptor binding domain head. They can also assume specific shapes along the glycoprotein neck and pre-fusion complexes, preventing conformational changes required for membrane fusion preceding entry [259]. Since the S protein of SARS is both a receptor binding (S1) and class I fusion protein (S1/2 and S2), there is potential for multiple antibodies to form and prevent SARS entry. 2003 SARS S-specific human monoclonal antibody CR3022 provides protection against the virus, and recent experiments revealed in vitro potent cross-neutralization against the SARS-CoV-2 S RBD [260]. Donated antibody serum against SARS-CoV-2 from survivors is currently being researched, but at the time of writing, data meant to establish its effectiveness are inconclusive. Regardless, donated plasma must come from healthy blood donors with known medical history having no evidence for past blood-borne infectious diseases [261].

Humoral immunity against 2003 SARS was dominated by IgG antibodies specific to S (residues 669–1255) and N [262]. In non-intensive care unit (ICU) patients, an increase of S-IgG positively correlated with a decrease in C-reactivity, a marker for patient recovery; it was the longest and most secreted antibody [263]. Despite these vulnerabilities, the S protein provides structural defense against potential antibodies through glycosylation. Anti-SARS-CoV-2-IgG antibodies appeared weeks before clearance of infection, suggesting that these antibodies were not neutralizing [264], and that patients must survive the infection long enough for true neutralizing antibodies to develop. It is possible that S glycosylation provides a glycan shield against antibodies, such as those made against the fusion protein of HIV-1 [265]. Glycan shields are characterized by densely clustered oligo-mannose glycans that extensively interact with each other as well as intricate structures within the protein to shield it from antibodies [266].

Glycan analysis revealed a preference for oligo-mannose type glycans in SARS and MERS as well as α- and δ-CoVs [266,267]. SARS-CoV-1 S contains 22 N-linked glycan sites, while SARS-CoV-2 S has 23, sharing 18 of its glycan sites with its “predecessor” [266]. However, Cryo-EM and further glycan analyses revealed that SARS and MERS mannooligosaccharides are more loosely scattered around the S1 head and S2 subunits and lack the characteristic dense organization of oligo-mannose when it serves as a glycan shield [266,268]. The S2 subunit, which forms a class one fusion complex, is much less susceptible to mutation than S1, due to less selective pressure and its conserved mechanical nature. In fact, bioinformatic analysis of SARS-CoV-2 and SARS-CoV-1 revealed that S2 glycan sites were completely conserved while S1 glycan sites experienced deletions and additions of other residues [266]. It is difficult to say whether the loosely clustered glycans in SARS and MERS provide a glycan shield, but extensive glycosylation of the protein has been confirmed.

Since antibodies preferentially target the S1 subunit [266], it is reasonable that movement of glycans, due to a higher mutational frequency in the S1 subunit, can provide structural differences sufficient to prevent cross immunity between previous and novel versions of the virus as the virus transits host reservoirs over time. Indeed, stripping the SARS S of N-glycans with peptide N-glycosidase-F abolished neutralization of the protein by purified antisera developed against purified virions [269], thus indicating the specificity of anti-S antibodies.

O-linked glycosylation of the SARS-CoV-2 S protein exists on Ser68, Thr323, Ser325, Ser673 and Thr678, with the last 3 being conserved O-glycosylation sites among CoVs [268,270]. Similar to the N-linked mannooligosaccharide shield, O-linked glycans can form glycan shields by forming mucin-like domains [270]. Although only a few O-linked glycan sites have been confirmed in SARS-CoV-2 S, data on this S protein are still preliminary because methodologies for the extraction of the monomeric/trimeric proteins, imaging, and computational predictions can affect results. Thus, the importance of glycosylation of SARS-CoV-2 S, and its role in immune evasion, have yet to be fully elucidated. Since glycans are important for the development of vaccines as antigens and adjuvants [271], the densely glycosylated S protein will likely play a substantial role in the search for vaccines.

Palmitoylation of S protein has both accessory and replicative functions. The S protein is palmitoylated on cytoplasmic cysteine clusters within the endo-domain [272]. Removal of all cytoplasmic palmitoylated cysteine residues does not affect folding, trafficking or association with M protein [273]. Intriguingly, deletion of the two cysteine residues C1234 and C1235 in SARS S prevented its incorporation into VLPs, despite still being able to associate with M [274]. Similar results were found for TGEV [275]. During receptor binding, palmitoylation promotes association with detergent-resistant membrane microdomains associated with ACE2 on cell membranes [218]. This hydrophobic nature of palmitoylated S improves binding with receptors associated with lipid rafts as well as syncytia formation [218]. A preprint study of SARS-CoV-2 S revealed 9 putative palmitoylation sites similar to SARS-CoV-1. SARS-CoV-2 S palmitoylation plays a key role in infecting cells with high cholesterol density in their membranes. During inflammation, certain cytokines induce production of cholesterol on cell surfaces, which subsequently promotes additional infections, thus upregulating further inflammation. Hence, cholesterol may be a determinant of pathogenicity in SARS-CoV-2 [276].

### 10.5. A Focus on M Protein PTMs

The M proteins of all known CoVs are O- and N-glycosylated which contribute to folding, structure, stability, trafficking and immune responses [277]. Glycosylation sites and hydropathy patterns are remarkably well conserved in CoV M proteins [278], suggesting an importance for function. SARS-CoV-1 M protein is glycosylated on residue N4, although the consequences of this carbohydrate derivative are elusive. Suppression of this N-glycosylation site does not impair its accumulation in the Golgi or the assembly and infectivity of SARS virions [279]. O-glycosylation in MHV and TGEV M proteins was found to induce antiviral cytokine IFN-α, and mutating O-glycan sites to N-glycan sites in MHV induced higher levels of IFNs [278,280]. Strangely, changing the glycosylation state of M to (O^−^/N^+^) in recombinant MHV improved infectivity in vivo in mice [280]. Despite these pathological observations, other roles for O- or N-glycosylation in M for CoVs remain unknown, as suppressing glycosylation did not hamper recombinant viral production [278].

### 10.6. A Focus on E Protein PTMs

The role of glycosylation in E protein structure, localization and stability is relatively under-studied [148]. SARS E is reported to have two putative glycan sites, N48 and N66, which may or may not be glycosylated in the fully processed protein [281]. Typically, SARS E has its HD facing the membranes, but whether the C terminal end faces the cytosol while N faces the ER lumen, or both terminal ends face the cytosol, is uncertain [281,282]. However, in at least one minor form, SARS E N66 is glycosylated with the C-terminal tail exposed to the lumen of the ER/Golgi [281]. This additional minor conformation with glycosylated N66 may contribute to alternative dimers and trimers. Since it is minimally glycosylated, it is unfortunately difficult to establish a role for singly glycosylated proteins such as E [251].

E protein has a cluster of 2–3 cysteine residues (SARS-CoV-1 C40, C43 and C44) on the carboxy side of the HD that are all palmitoylated in IBV and SARS [149,283]. While these cysteines may participate in disulfide bridges to form homodimer/trimers [284] and other possible hetero-oligomers, these residues are not important for homopentamer formation or IC activity [283]. Mutating these residues to alanine prevents E from oligomerizing with M, but not with N [149,283]. MHV E palmitoylation on C40, C44 and C47 likely promotes association with membranes, possibly embedding part of its α-helical HD into the membrane while the palmitoylated cysteine cluster stabilizes its association with membrane lipids [257]. This interaction may contribute to the production of viral particles, as mutating the cysteines dramatically reduces the production of VLPs [257]. These findings support the hypothesis that multiple conformations of E play different roles of CoV replication and pathogenesis. In addition, these different conformations and palmitoyl-assisted membrane anchoring may contribute to viral particle structural integrity.

PL^pro^ contains secondary deubiquitinating activity, suggesting a role in host and viral protein modulatory function. Despite this property, the E protein is ubiquitinated. Following the theme of viral protein ubiquitination, E ubiquitination may allow avoidance of host cell defense mechanisms. The N-terminal Ubl1 domain of nsp3 interacts with E, and the complex localizes in the cytoplasm of infected cells [150]. As stated earlier, the nsp3 Ubl1 domain can act as an anchor for other viral proteins such as N. Since nsp3 and E are involved in viral replication, this association could be important for the synthesis of new virions, bringing E close to the RVN. Additionally, nsp3 deubiquitinating PL^pro^ may dynamically alter the ubiquitination state of E, regulating potentially different protein-protein interactions, protection from the proteasome, and sorting of the protein. Cellular ion channels are also known to be regulated by ubiquitination, where misfolded membrane-bound protein is marked for degradation [285]. E has many conformations and oligomerization states, and perhaps, the concentrations of each may be influenced by ubiquitination. While it is not clear what the role of E ubiquitination is, suppressing ubiquitination in MHV interferes with viral RNA synthesis and may inhibit proteasome and viral nsp proteolytic activities [286].

## 11. Viral Responses to and Interference with Normal Cellular Function

### 11.1. Interference with Host Immunological Responses by Interferon (IFN) Antagonism

The type-I IFN system is an important first line of defense against viral infections, participating specifically in antiviral responses. IFN is an effective inhibitor of coronavirus replication and is detected in significant amounts in CoV infected animals and cell lines, but its expression is delayed both in vivo and in vitro [287,288]. In fact, SARS-CoV-2 is more susceptible to IFN treatment than its predecessor, despite its strong ability to suppress IFN pathways [289]. Activation of IFN pathways can occur through detection of dsRNAs through cytosolic RIG-I and RIG-I-like receptors (RLRs) such as melanoma differentiation gene 5 (MDA5) [290]. Toll-like receptors (TLRs), TLR3, TLR7 and TLR8, also detect viral single or double stranded RNAs [291]. Upon activation, RIG-I and MDA5 caspase activation recruitment domains (CARD) are modified with ubiquitin [292] and bind with adaptor mitochondrial antiviral signaling (MAVS) protein, also known as IFN-β promoter stimulator 1 (IPS-1), to form the IPS-1 signalosome [293]. The IPS-1 signalosome then interacts with IKK-related kinases, TANK-binding kinase 1 (TBK1) and IκB kinase (IKKε) [293]. Both kinases can phosphorylate interferon regulatory factors 3 and 7 (IRF3/7) [290]. Phosphorylated IRF3 and IRF7 form homo- and heterodimers and translocate to the nucleus to activate expression of IFN-α/β [290]. The IPS-1 signalosome can also recruit IKKα and IKKβ kinases which activate NFκB. NFκB then translocates to the nucleus to activate expression of proinflammatory cytokines, TNFα and IL-1β, and it upregulates type-I IFN expression [294]. Similarly, TLRs recruit TRIF and/or MyD88 which activates IKKε/IKKi kinases which phosphorylate IRF-3 [291]. TLRs can also activate NFκB through MyD88-IRAK-TRAF6 signaling which activates IKKα/IKKβ [291]. Once IFNs are secreted, they behave as autocrine and paracrine factors to stimulate the expression of IFN-stimulated genes (ISGs) through Janus activated kinase (JAK)-signal transducers and STAT signaling pathways [295]. Binding of IFN to IFN receptors on cell surfaces stimulates the JAK-STAT pathway, which utilizes JAK1 and Tyk2 kinases to phosphorylate STAT1 and STAT2 which triggers their dimerization and translocation to the nucleus where they activate ISGs [296]. Downstream effects of ISGs include upregulation of chemokines (including additional IFNs) and chemokine receptors [297], induced resistance to viral replication in cells [298], activation of monocytes/macrophages [299], activation of Natural Killer cells to kill virus-infected cells [300], and regulation of adaptive T and B cell responses [301,302]. The antiviral innate immune responses of animal hosts commonly interfere with essential viral processes such as the formation of replication-associated membrane structures [303]. In response to the anti-viral activities of these host proteins, many viruses combat the interferon-mediated anti-viral activities of the host by a number of mechanisms [304]. Coronaviruses are equipped with a large array of viral proteins that have secondary functions in IFN suppression or evasion including nsps 1, 3, 7, 12, 13, 14, 15, 16 [305,306,307,308], structural proteins M [309], N [292] and E, and accessory proteins ORF3b, 4a [310], 4b [311], 5 [292], 6 [306,307] and 9b [292].

### 11.2. The M Protein

M-proteins have been reported to be potent interferon (IFN) antagonists in MERS and SARS-CoV-1/SARS-CoV-2 [306,308,309]. In these experiments, the genes were individually cloned into plasmids, transfected into cells, and expressed. While all three of these proteins were effective, ORF4a seemed to be the most potent at counteracting the antiviral effects of IFN via the inhibition of IFN-β promoter activity and NF-κB activation as well as the ISRE (interferon-stimulated response element) promoter signaling pathways [309]. These studies were continued with SARS-CoV-1 M protein, showing that M suppresses type I interferon production by impeding the formation of a functional TRAF3-containing complex. This IFN antagonizing activity is mediated by the first TMS (TMS1) at the N-terminus of the protein. Some specificity was surprisingly noted, since the human Coronavirus HKU1 M protein lacked the inhibitory activity observed for the SARS-CoV-1 M. TMS1 of SARS-CoV-1 M targets the protein to the Golgi apparatus, and Golgi localization seems to play a role in its action as an IFN antagonist. Using the MERS-CoV M protein, the authors suggested that TMS1 prevents the interaction of TRAF3 with its downstream effectors [312,313], confirming its ability to help evade the host innate antiviral response by suppressing type I IFN expression in response to various agents and RNAs. They reported that M interacted with TRAF3, blocking the TRAF3-TBK1 association, which in turn reduced activation of the INF regulatory factor 3 (IRF3). Liu et al. also found that the N-terminal hydrophobic TMS, but not the C-terminal hydrophilic region of M, was important for the response, confirming the results of Siu et al. [312]. The M-mediated interferon antagonism noted here seems to be a common characteristic of a large number of (but definitely not all) viruses from different viral classifications [314,315].

Surprisingly, M may also be able to promote IFN-β induction via a Toll-like Receptor (TLR)-related, TRAF3-independent mechanism [316]. In this case, M itself (rather than its mRNA) seemed to function as the cytosolic pathogen-associated molecular pattern (PAMP) to stimulate type I interferon production. In fact, both NF-κB and TBK1-IRF3 signaling cascades were reported to be activated by M-gene products. Activation of IFN-β production seemed to be generated from within the cell, and the wild type M-protein induced production of both IFN-β and NFκB through a TLR-related signaling cascade. Interestingly, a V68A mutant of M had the opposite effect, markedly inhibiting SARS-CoV-promoted INF-β production [316]. These observations illustrate the complexity of virus-host cell interactions and reveal the high degree of specificity observed for different envelop viruses.

### 11.3. The N Protein

The N-protein of the MERS coronavirus suppresses Type I and Type III interferon (IFN) induction (virus-induced IFN-β and IFN-lambda1) by targeting RIG-I signaling. This is accomplished by reducing the IFN gene promoter activities and therefore their mRNA levels, thereby blocking production of the bioactive IFNs. The C-terminal domain of the N-protein plays a pivotal role in this antagonistic activity, and it is particularly important, as these interferons are at the frontline of the larger antiviral defense that triggers the activation of hundreds of downstream antiviral genes [317].

Details of the transcriptional signaling pathway have been elucidated [317], and in an earlier study, Likai et al. [318] found that the porcine δ-coronavirus N-protein suppressed IFN-β production in piglets. These observations suggest that in many, if not all coronaviruses, the N-protein functions to allow the virus to escape the immune surveillance of the host. In all studied cases, the mechanism of suppression involves the N-protein targeting the promoters of interferon genes. This is accomplished by targeting the retinoic acid-inducible gene 1 (pRIG-1) and the TNF receptor by direct interaction. The two studies, using very different coronaviruses, indicate that similar mechanisms of action are involved in both cases. In fact, even earlier studies had provided evidence for such a mechanism [319].

SARS N protein also interferes with TRIM25-mediated RIG-I ubiquitination [292]. Rather than binding to RIG-I or MDA5 [320], N protein associates with the RIG-I effector molecule, TRIM25 [292]. Upon detecting a PAMP, RIG-I is ubiquitinated by TRIM25 [321] to begin the essential antiviral signal cascade. SARS N protein C-terminal residues 364–422 competitively bind to the TRIM25 SPRY domain and interferes with its binding to RIG-I, disrupting the necessary ubiquitination [292]. Such inhibition would blind the cell from ever detecting the presence of non-host RNAs accumulating during viral replication through RIG-I. Downstream IFN signaling is also disrupted by the SARS-CoV-2 N protein by inhibiting the phosphorylation of STAT1 and STAT2 through direct binding to STATs. Coimmunoprecipitation assays revealed that truncated N residues 1–361 are sufficient to prevent STAT signaling, with region 319–422 aas being indispensable for STAT binding [322].

## 12. Nonstructural Protein Interference with IFN Gene Expression

### 12.1. nsp1

Nsp1 suppresses IFN activation in an unprecedented manner, unique to all other enveloped RNA viruses considered thus far. Rather than inhibiting protein interactions involved in IFN cascades, nsp1 promotes the degradation of host mRNAs. To assess the contribution nsp1 has to IFN suppression through host RNA degradation, SARS-CoV-1 carrying mutant nsp1 had higher levels of IFN-β coupled with higher levels of host mRNAs. Specifically, residues 160–173 in the C-terminal end of nsp1 participate in mRNA degradation [323]. SARS-CoV-1 nsp1 localizes to translation complexes and has been shown to directly bind to the 40S ribosomal subunit to access the mRNAs, abrogating translation [324]. In MERS, an endonuclease was confirmed to exist within nsp1 despite not being able to bind to the 40S ribosome, indicating that RNA degradation activity may vary even within the β-CoVs [325]. SARS-CoV-2 nsp1 was shown to bind to both 40S and 80S ribosomal subunits through its C-terminal region, physically blocking RNAs from entering the entrance region of ribosomes. It was proposed that SARS-CoV-1 nsp1 may degrade host mRNAs in a two-pronged manner, where it first binds to 40S ribosome subunits, and then applies modifications to host RNAs at the 5′ caps, rendering them translationally incompetent [326]. It was suspected that nsp1 triggers template-dependent endonucleolytic RNA cleavage in the 5′ region of RNAs [327], which is then completed by exonucleolytic activity from host Xrn1 [324]. Viral transcripts have been reported to possibly escape nsp1-induced degradation due to the differences in 5′ caps in viral and host transcripts [327]. When viral proteins are expressed on clonal plasmids, nsp1 can promote the degradation of its own transcripts in transfected cells [323]. Nsp1 does not prevent IRF3 dimerization but does prevent the expression of IFN transcripts. In addition to its endonucleolytic activity, nsp1 can disrupt downstream IFN signaling, where SARS-CoV-1 nsp1 inhibits STAT1, but not STAT2 phosphorylation [328].

### 12.2. nsp3

As noted previously, nsp3 is a large nonstructural protein containing multiple domains, and it participates in a wide array of functions. Its PL^pro^ domains have deubiquitinating and deISGylating activities and are speculated to participate in immunomodulation. Indeed, SARS and the HCoV-NL63 PL2^pro^ domains can interfere with IRF-3 phosphorylation without affecting respective kinases and stimulation of NFκB dependent genes [329,330,331]. Interestingly, the enzymatic activities of nsp3 are not solely responsible for IFN-β suppression. Deleting the catalytic residue, C1678 in SARS-CoV-1, and H1836 in HCoV-NL63 PL2^pro^, to eliminate proteolytic and deubiquitinating activities of the protein, only slightly decreased IFN suppression [331]. Treating PL2^pro^ transfected cells with protease inhibitor GRL-0617S had no effect on IFN suppression but did abrogate NFκB stimulated gene suppression [331]. Since nsp3 is a membrane-spanning protein, the authors of this study also examined if TM forms of the PL2^pro^ domain could inhibit IFN expression. Truncating nsp3 to only include the PL2^pro^ domain attached to a TMS was shown to be a potent IFN antagonist, and it could suppress N-RIG stimulated IFN-β production [331].

To investigate the possible role deubiquitination has on IFN suppression, IRF-3(5D), a phosphomimetic of IRF3, was shown to be deubiquitinated by PL2^pro^. Despite being deubiquitinated, IRF-3(5D) was still able to dimerize, translocate to the nucleus and bind to DNA, but it could not induce IFN expression. The authors proposed that its interaction with other transcriptional machinery is altered so that IFN expression cannot be achieved [332]. These results suggest that nsp3 is a potent inhibitor of IFN expression both upstream and downstream of IRF-3 phosphorylation. It has yet to be detailed how nsp3 can prevent the phosphorylation of IRF-3. Enzymatic activity may also be required for NFκB stimulated gene suppression [331].

## 13. Accessory Protein Interference with IFN Expression

### 13.1. ORF6

The SARS-CoV-1 ORF6 protein has been shown to have IFN-inhibiting abilities, suppressing both upstream and downstream effectors of IFN pathways. Expression of the protein suppressed Sendai virus-induced IFN expression by inhibiting phosphorylation and subsequent translocation of IRF3 to the nucleus [308,333]. ORF6 was also shown to inhibit STAT1 nuclear translocation, despite not preventing STAT1 phosphorylation [334]. Similarly, SARS-CoV-2 ORF6 is able to broadly suppress type-I IFN expression in vitro. Clonal expression of SARS-CoV-2 ORF6 and C-terminally truncated ORF6 inhibited multiple stages of IFN activation as well as downstream pathways of IFN signaling. Specifically, residues 53–61 of the protein overexpressed in HEK293T cells suppressed IRF3 activation by interfering with RIG-I, MDA5, and MAVS complex assembly [308]. Interestingly, the same region was also able to inhibit STAT1 nuclear translocation in IFN-stimulated HEK293T cells [308]. A possible explanation is that SARS-CoV-1 ORF6 localizes in the ER/Golgi membranes. Its C-terminus binds to nuclear import factors karyopherin-α2 and karyopherin-β1, disrupting the formation of nuclear import complexes. Phosphorylated STAT1 is then unable to enter the nucleus. Deletion of ORF6, or removal of the C-terminus, restored STAT1 nuclear translocation [335]. It therefore seems that ORF6 can prevent the expression and secretion of IFNs, thus preventing the downstream upregulation of ISGs.

### 13.2. ORF3b

Lone transfection of clonal SARS-CoV-1 ORF3b in A549 cells co-infected with recombinant Newcastle disease virus (NDV) prevented replication in the presence of type-I IFN-rescued NDV replication. ORF3b is able to prevent IRF-3 phosphorylation, and thus its translocation to the nucleus. Interestingly, ORF3b was found to localize to the nucleus and nucleolus of cells, associating with B23, C23, and fibrillarin through a nuclear localization signal (NLS) in its C-terminal end [336]. Despite its nuclear localization, it is the cytosolic ORF3b that participates in IFN antagonism, as recently shown for SARS-CoV-2. In fact, deletion of the NLS improves the IFN antagonism of SARS-CoV-2 variants, making this region of the protein an indicator for coronaviral pathogenesis [337]. Similar to ORF6, ORF3b also prevents stimulation of downstream IFN pathways, inhibiting expression from an IRSE promoter [334]. The function of nuclear localization of ORF3b has yet to be detailed.

## 14. Complement Activation by CoV Structural Proteins

S, E, N and a few nsps likely play roles in the activation of complement, the immediate innate immune response and the bridge between innate and adaptive immune systems. Complement is a double-edged sword and has only recently undergone more thorough investigation as a major contributor to over-inflammation and pathology. Progression of disease in many pathogenic infections are often the result of hyperactive innate immune responses, inducing severe inflammation. Complement activation is a multistage process involving a large array of activation products. It is a crucial driver of early inflammation and provides protection from infections, stimulating proinflammatory and cytotoxic cytokine secretion, activation and proliferation of leukocytes, vascular constriction, and stimulation of adaptive immune cells (B and T cells) [338,339]. If complement is overstimulated, a cytokine storm may ensue, and disease can be characterized by intense fever, immense vasoconstriction, plasma coagulation, necrosis of infected cells and severe tissue damage. SARS-CoV-1 and other respiratory viruses such as Influenza induce intense fever, severe pulmonary tissue damage, vasoconstriction, and thrombosis in alignment with symptoms of overactivated complement. Evidence exists supporting the suggestion that MERS, SARS-CoV-1 and SARS-CoV-2 all induce complement. Mice infected with either SARS-CoV-1 or MERS have elevated levels of complement proteins in sera [228], and preprint studies on SARS-CoV-2 patients revealed elevated complement-associated proteins in alveolar spaces and blood vessels [229]. Newer proposed treatments for viral infections involve suppressing complement to increase the host tolerance for the pathogen, allowing the virus to proliferate while reducing the severity of pathogenicity.

Complement can be activated via three routes, first, the classical pathway, mediated by natural IgM or antigen-specific IgG, second, the mannose binding lectin (MBL) pathway, mediated by MBL binding to antigen, and third, the alternative pathway, activated by plasma. In all three pathways, production of Complement (C)3 cleavage products, C3a and C3b, are required to begin the downstream effects. In the classical pathway, pentameric IgM or at least 2 IgGs bind(s) to antigen and associate(s) with complement C1 proteins, C1q, C1r and C1s, to form the C1-complex. The C1 complex activates the C1r subunit, a serine protease which splits C4 and C2 into C4a plus C4b and C2a plus C2b, respectively. C4b and C2a associate to form C4bC2a, the C3-convertase which cleaves C3 into C3a and C3b. Similarly, the MBL pathway utilizes opsonin, MBL and ficolins to activate MBL-associated serine proteases (MASP-1 and MASP-2) which cleave C2 and C4 into C2a plus C2b, and C4a plus C4b, respectively. The alternative pathway differs the most and is independent of the C4 derived protease. Rather, it requires the spontaneous hydrolysis of C3 in plasma to form C3(H_2_O). C3(H_2_O) binds to factor B (fB) to form C3(H_2_O)fB, which is cleaved by factor D (fD) to form the alternative fluid phase C3 convertase C3(H_2_O)Bb which can cleave C3 into C3a and C3b. This spontaneous production of C3(H_2_O)Bb ensures a stable and abundant level of C3b in plasma. C3b deposits on pathogens or infected cell membrane surfaces. Free C3b can induce the alternative pathway if it directly binds to the surface of a pathogen. Membrane bound C3b is still able to associate with fB, and in the presence of factor D, it will produce membrane-bound C3bBb, the alternative pathway C3 convertase. All complement pathways converge on the C3bBb C3 convertase to promote the cleavage of C3 in a positive feedback loop. C3a acts as a proinflammatory chemokine. Downstream, C3b becomes a C5 convertase by associating with other C cleavage products, the classical/MBL (C4b2b3b or C4b2a3b), or an alternative (C3bBbC3b). Terminal C5 cleavage products, C5a and C5b, result in a final form of complement. C5a, like C3a acts as a chemoattractant for leukocytes. C5b can bind to cell surfaces and oligomerizes with C6, C7, C8 and poly C9 to produce the C5b9 membrane attack complex (MAC), the innate immune system’s cytotoxic warhead. MAC breaches a hole in bacteria, virus-infected (recognized non-self) cells, and even viral envelopes, causing extracellular fluids to rush into the cell/virus, inducing lysis. Other roles of C5b and C5b9 promote chemokine secretion and inflammation [338,339] (see Figure 5 for an illustration of the complement pathways). A majority of complement proteins are produced in the liver and secreted into the blood [340]. Damaged liver tissue seems to require activation of complement for regeneration, relying specifically on C3 and C5 [340], complicating the balance between suppressing and activating complement in diseases that affect the liver. Extensive damage in the liver has been linked to severe disease in SARS-CoV-2 infections [341].

Inhibition of complement pathways attenuate disease progression despite continued replication of the pathogen in the host. Inhibiting C5 reduces intravascular coagulation and prevents organ failure, cytokine storms and sepsis in *E. coli*-infected Baboons [342]. For influenza virus, C5 induces over-recruitment of neutrophils and CD8+ T cells as well as cytokine secretion, inducing acute lung injury in H1N1 or H5N1 infected mice. Treatment with a C5 inhibitor significantly attenuates respiratory inflammation and tissue damage [343]. Along with septic shock, typically, multi-organ failure and kidney damage are associated with complement overactivation [344,345]. Since severe SARS-CoV-1 infections accompanying kidney damage, while rarer, are linked to systemic over inflammation rather than viral tropism for this tissue [346]. It is not unlikely that complement plays a role in the multiple organ failure seen in SARS-CoV-2. Investigations into the role of complement in CoV-induced disease revealed that SARS-CoV-MA15 infected C3-/- mice resisted severe disease progression. In comparison, control mice, having elevated complement proteins in mouse sera, exhibited 15% weight loss with lung tissue damage [228]. C3-/- infected mice did not lose weight 2–4 dpi, had reduced cytokine proinflammatory secretions (IL-6, TNF-α, and IL-1β), had reduced monocyte infiltration and exhibited little pulmonary tissue damage. It is likely that multiple branches of complement activation are required for infection in SARS-infected mice, as neither C4-/- nor fB -/- mice were protected from weight loss [228].

Complete suppression of C3 in humans is probably not a valid strategy for combating SARS-CoV-2 due to the necessity of C3 in other immune pathways. Despite having a seemingly beneficial effect in SARS infected mice, C3-/- mice infected with H5N1 or H1N1 actually had more inflammation and tissue damage due to failure to activate adaptive humoral and cell immunity [347,348,349]. Rather, downstream C5 products are responsible for severe and lethal infections similar to the *E. coli* and influenza studies [229]. Endothelial injury from C5 activation products was detected in infected and damaged ACE2^+^ tissues. Together with the formation of C5 products, over recruitment of neutrophils and macrophages was observed. C5a interacts with membrane C5aR on endothelial cells, inducing downregulation of thrombomodulin and activation of coagulation with secretion of P-selecting promoting platelet adhesion, aggregation and recruitment of white blood cells. Besides forming MAC, C5b9 induces endothelial activation and dysfunction, upregulating tissue factors and adhesion molecules for migrating white blood cells. Additional inflammatory chemokines are secreted along with increased vascular permeability and coagulation. In unpublished observations, abundant C5b9 was observed in microvasculature of interalveolar septa, large caliber vessels of the lung parenchyma and microvasculature in occluded arteries of SARS-CoV-2 patients [229]. C5b9 deposits were also found in septal capillaries colocalized with the S and E proteins, indicating that CoV structural proteins are involved in the induction of complement. Downstream suppression of C5a and C5b activities would be reasonable as all complement pathways result in C5 cleavage products. This would prevent the most severe effects of complement from occurring without affecting other peripheral pathways, such as stimulation of adaptive immunity [339]. Anti C5aR antibodies prevented MERS-induced upregulation of proinflammatory cytokines in serum, thus reducing leukocyte infiltration and tissue damage [350]. Suppression of C5 products could be achieved by the use of the approved drug, eculizumab, which inhibits C5, preventing its cleavage [351], or by candidate C5aR inhibitor, CCX168, currently in phase III clinical trials [352].

## 15. Induction of Endoplasmic Reticulum (ER) Stress and the Unfolded Protein Response (UPR)

Perturbation of the ER, for example by pore-formation, causes ER stress, leading to the activation of cell signaling pathways including the unfolded protein response (UPR). As noted above, SARS-CoV-1 uses the ER/Golgi apparatus for synthesis and processing of viral proteins, and for this purpose, it uses the UPR. Although several viral proteins may contribute to the UPR, the Spike (S)-protein appears to be the primary inducer of several UPR effectors, including glucose-regulated protein 78 (GRP78), GRP94, and the C/EBP homologous protein. However, the expression of S exerts different effects on the three major signaling pathways of the UPR. Thus, it induces GRP78/94 through the PKR-like ER kinase, PERK, but it has no effect on activating transcription factor 6 or X box-binding protein 1. The S-protein appears to specifically modulate the UPR to facilitate viral replication [176,353]. However, overexpression of ORF3a, ORF3b, ORF6 or ORF7a can also induce apoptosis. Interestingly, inhibitors of Caspase-3 and JNK block ORF-6 induced apoptosis. Thus, ORF-6 induces apoptosis via Caspase-3-mediated ER stress and JNK-dependent pathways [175]. ORF3a also down regulates the type 1 interferon receptor [354], while Nsp6 activates omegasome and autophagosome formation [355]. Interestingly, the E-protein of SARS-CoV-1 seems to decrease the stress responses while increasing inflammation [179], yet the same protein, as well as N of Porcine Epidemic Diarrhea Virus (PEDV), can cause ER stress. However, both proteins also up-regulate interleukin-8 expression [178,356,357], while overexpression of Nsp7 down-regulates interleukin 8 [356]. In fact, many of the CoV proteins, including 3, 8b, and the ion channel activity of the IBV E-protein, influence ER stress and the translation apparatus [24,82,356,358]. Interestingly, although not essential for replication, glycosylation of the IBV M protein ectodomain plays important roles in activating ER stress, apoptosis and the pro-inflammatory response, thereby contributing to the pathogenesis of IBV [359]. All of these analyses reveal (1) how complicated the viral induction of ER stress is, and (2) the large number of viral proteins that influence this process.

## 16. Coronavirus-Induced Host Cell Cycle Arrest

Several early studies demonstrated that various proteins encoded within coronaviral genomes can cause cell cycle arrest in the infected cells in various growth phases. One of these is 3a of SARS-CoV-1 which is mainly localized to the Golgi apparatus together with M in co-transfected cells. Expression of 3a inhibited cell growth and prevented 5-bromodeoxyuridine incorporation, suggesting that 3a deregulates cell cycle progression [360]. 3a expression blocked cell cycle progression at the G1 phase in various tissue cells 24–60 h after transfection. Mutational analysis of 3a revealed that the C-terminal region, from residue 176, which includes a potential calcium ATPase motif, was essential for cell cycle arrest. As noted above, like the M-protein, 3a predominantly localized to the Golgi apparatus, with its N-terminus residing in the lumen and its C-terminus in the cytosol. In the relevant experiments, 3a expression correlated with a reduction of the cyclin D3 level. Increases in p53 phosphorylation on Ser-15 were observed in both SARS-CoV-1 M and 3a transfected cells, suggesting that this phosphorylation activity might not be responsible for the 3a-induced G0/G1 phase arrest. Thus, there was evidence that 3a and M might function independently to inhibit cell cycle progression, but that their detailed mechanisms might be different. ORF7a expression may also block cell cycle progression in the G0/G1 phase, and it apparently can induce apoptosis via a caspase-dependent pathway [361]. ORF7a expression is associated with blockage of cell cycle progression in several cell lines after 24 to 60 h post-transfection. Mutational analysis of ORF7a revealed that the domain spanning amino acyl residues 44–82 was essential for its induction of cell cycle arrest. Since ORF7a expression correlated with a reduction of cyclin D3 mRNA levels and phosphorylation of the retinoblastoma (Rb) protein on serine residues, it was suggested that the insufficient expression of cyclin D3 might have caused the decreased activity of cyclin D/cdk4/6, resulting in the inhibition of Rb phosphorylation. Accumulation of hypo- or non-phosphorylated Rb thus may have prevented cell cycle progression during the G0/G1 phase.

Virulent strains of porcine epidemic diarrhea virus (PEDV), an enteropathogenic α-coronavirus, cause a highly contagious enteric disease in swine, characterized by severe enteritis, vomiting, and watery diarrhea. Xu et al. [362] investigated the subcellular localization and function of the PEDV M-protein through examination of its effects on cell growth, cell cycle progression, and interleukin 8 (IL-8) production. Their results revealed that after infection, the M-protein seemed to localize throughout the cell cytoplasm. M altered porcine intestinal epithelial cell line (IEC) growth, and it induced cell cycle arrest at the S-phase via the cyclin A pathway. S-phase arrest proved to be associated with a decreased level of cyclin A, but M did not induce endoplasmic reticulum (ER) stress (see the next section). Moreover, it did not activate NF-κB which is important for IL-8 and Bcl-2 expression. Thus, the PEDV M-protein induces cell cycle arrest when cells are in the S-phase. Sun et al., 2018 [363] confirmed many of the observations of Xu et al. (2015) [362], and further showed that the p53-dependent pathway plays an important role in PEDV-induced cell cycle arrest. In fact, inhibition of p53 signaling reversed arrest. They additionally showed that cell cycle arrest contributes to viral infection and involves down-regulation of the Cyclin E protein gene.

## 17. Coronavirus-Induced Autophagy and Abortive Apoptosis

Macro-autophagy (hereafter referred to as autophagy) is an evolutionarily conserved intracellular catabolic transport route that generally allows the lysosomal degradation of cytoplasmic components, including bulk cytosol, protein aggregates, damaged or superfluous organelles and invading microbes [356]. Notably, autophagy participates in both innate and adaptive immune pathways. The innate role is through an autophagy subroutine called xenophagy for the elimination of intracellular parasites and viruses. The adaptive immune system utilizes autophagy for the purpose of antigen presentation. Autophagy allows for cells to cross-present antigens between the MHC class I and MHC class II molecules. Typically, MHC class I present antigens of endogenous sources, while MHC class II presents antigens from extracellular spaces. Autophagy permits endogenous antigens to enter the MHC class II presentation pathway. It is then no surprise that several viruses have evolved mechanisms to inhibit or hijack autophagy pathways and associated proteins. It remains a debated question if Coronaviruses can also utilize autophagy for their own replication, or if autophagy is an effective antiviral response to Coronavirus infection.

Porcine hemagglutinating encephalomyelitis virus (PHEV) infection induces atypical autophagy and causes the appearance of autophagosomes, but it blocks fusion with lysosomes [360]. In addition, transmissible gastroenteritis virus (TGEV) infection induces autophagy of mitochondria (mitophagy) to promote cell survival and possibly viral infection while counteracting oxidative stress and apoptosis [361]. In fact, non-canonical autophagy is believed to converge with the infection cycles of many DNA and RNA viruses that utilize membranes from the ER and cis-Golgi [362]. PL2^pro^ may act as a novel autophagy-inducing protein, but it induces incomplete autophagy by increasing the accumulation of autophagosomes while blocking the fusion of autophagosomes with lysosomes. Furthermore, PL2^pro^ interacts with the key host cell autophagy regulators, LC3 and Beclin1 to promote a Beclin1 interaction with STING, the key regulator for antiviral IFN signaling. Finally, knockdown of Beclin1 partially reversed the PL2^pro^ inhibitory effect on innate immunity while resulting in decreased coronaviral replication [364]. Nsp6 of β-CoVs MERS, SARS-1 and SARS-2, and γ-CoV IBV have also been documented to restrict autophagosome expansion, ultimately preventing the delivery of viral components to lysosomes for degradation [99]. While coronavirus replication complex formation requires constituents of the host autophagy system [365], it does not require the autophagy protein, ATG5, that normally completes autophagy and promotes fusion of the autophagocytic vesicles with lysosomes [102]. So far, there is little or no evidence that the M-protein plays more than an indirect role in autophagy.

Abortive apoptosis is a last resort mechanism of cells in response to intracellular stress, and detection of DNA damage. Many viruses can induce apoptosis in cells either “intentionally” for replicative purposes, or “unintentionally” due to consequences of hijacking cellular machinery. Whether or not apoptosis is beneficial or not to CoV replication remains questionable. While all CoVs induce ER stress, and can induce apoptosis, there are many proteins that suppress the UPR mediated abortive apoptosis pathway. Coronaviruses have been indicated to induce said intracellular stress as mentioned in earlier sections of this review, in particular the production of CoV proteins in the ER activates the UPR pathway. Specifically, the extensive post translational modifications of the various membrane spanning proteins in the CoV proteome rely heavily on the limited protein chaperons inside the ER [358]. Prolonged UPR and failure to reattain homeostasis leads to ER stress-associated abortive apoptosis [358]. The protein, ER-resident transmembrane kinase-endoribonuclease inositol-requiring enzyme 1 (IRE1), a UPR signal transduction molecule that behaves as a timer for heavy ER stress, indicates the cell to switch from cytoprotective phase to apoptosis. IRE1 functions as a RNase, splicing the mRNA of the X box binding protein 1 (XBP1) gene, producing XBP1s mRNAs which encode a potent activator of many UPR genes. Conversely, unspliced XBP1 confers an inhibitor of UPR genes. Thus, prolonged IRE1 signaling and splicing of XBP1 results in overactivation of UPR and decreased cell survival over time. MHV and IBV were shown to activate the IRE1-XBP1, but XBP1s protein expression is suppressed in MHV possibly by persistent phosphorylation of eIF2α, suppressing host translation [366]. Interestingly, SARS seems to prevent splicing of XBP1 altogether through some unknown mechanism related to the E protein [179]. Since CoVs rely on budding of virions from the host, as opposed to lysis, apoptosis would appear to be an inhibitory mechanism to optimal CoV replication.

## 18. Structural Proteins as Protective Antigens in Survivors, and Vaccine Development

### 18.1. S Protein as a Protective Antigen

Antisera of SARS survivors have shown representative IgG antigen recognition against the S1 subunit of the S protein [262,367]. Specifically, the RBD of the S1 subunit has been a prime target for adaptive humoral immunity against the virus [367]. Cryo-EM of the highly potent anti-RBD S230 antigen-binding fragment (Fab), purified from a SARS survivor antiserum, bound to the S protein, and displayed specific localization with the S^B^ domain existing in 2 states. The state 1 complex showed multiple orientations of each of the S230 Fabs associated with intermediate and open conformations of the S^B^ domain. State 2 complexes had all three S^B^ domains in the open conformation but lacked 3-fold symmetry [367], suggesting that S230 can bind to S^B^ domains in varying degrees of openness. The residues involved in the Fab-S^B^ complex were as follows: S230:CDRH2 F59 and S230:CDRH3 Y106, F107 and Y110, localized near SARS Y408, Y492, F460 and Y475 centered around L443 [367]. S230 potency may derive from its ability to mimic the ACE2 receptor and bind to the RBD with even higher affinity. Its mimicry of ACE2 also allows the molecule to trick S protein fusion activation, locking all S^B^ domains into the open conformation upon binding, leading to the relaxation of S2 subunit folds and subsequent proteolysis and premature activation [367]. Thus, S230 not only sequesters the S protein, but also deactivates its ability to fuse viral and host membranes. Because of the similarities between SARS-CoV-1 S and SARS-CoV-2 S, it is possible that cross immunity from polyclonal antibodies may exist between the two viruses. Consistent data have yet to confirm this suggestion, but both human and rabbit mono/polyclonal anti-SARS-CoV-1 S antibodies unfortunately had weak to no neutralizing capacity against either SARS-CoV-2 S pseudo-virions [32] or SARS-CoV-2 S itself [195], suggesting limited cross-immunity.

Since natural immunity against SARS-CoV-1 S is characterized by antibodies targeting the RBD, vaccination efforts have homed in on methods to develop anti-RBD vaccines. Successful and potent anti-RBD vaccines can be produced through recombinant IgG1-Fc-RBD_(318–510)_ in 293T cells [364] or a truncated S RBD_(318–510)_ fragment in mammalian 293T cells, insect Sf9 cells, or *E. coli* [365]. In these studies, potent SARS neutralizing antibodies were produced in rodent models, preventing infections both in vitro and in vivo. The strongest SARS neutralizing antibodies were IgGs from mice vaccinated with truncated RBD_(318–510)_, originating from transfected mammalian 293T cells, neutralizing 100% of SARS virions upon the first boost in veroE6 cells [365]. No viral RNA was detected in the lungs of mice 5 dpi, vaccinated with any of the truncated RBD vaccines, while unvaccinated mice suffered infection [365]. Regardless of the source, all RBD vaccines elicited strong anti-SARS activity, although it is not clear if the mechanism of protection is similar to that of S230. 

T cell responses, which are essential for the clearance of any viral infection, are also targets for activation by vaccines. Patients recovering from SARS have elevated levels of activated T cells. In a study measuring the adaptive immunity against SARS, at least 50% of SARS survivors tested positive for the T cell response a year after infection [368]. Patients who experienced severe illness had many memory T cells (CD26+/CD45RO+) and polyfunctional CD4+ T producing IFN-γ, TNF-α and CD107a degranulation. Many of the CD4+ T cells were largely specific for S protein [368]. In nearly all patients, the elevated T cell response was coupled with anti-S IgG antibodies, indicating that clearance of SARS is both humoral and cell-mediated, and centered around structural proteins, specifically S, although N is also targeted extensively. Thus, the development of vaccines that can also induce T cell responses would provide stronger protection, similar to immunity gained through infection.

DNA (or RNA) vaccination is a radically new method of vaccination. Viral DNA is cloned as cDNA plasmids and injected directly into a tissue of the subject in order to induce an immune response. DNA vaccines were shown to be effective against HIV, hepatitis B, hepatitis C, influenza and rabies [369]. Several studies involving developing cDNA vaccines based on the SARS S protein were able to rapidly mount humoral and cell-mediated immunity against the virus in rodent models. In a study by Huang et al., a full-length cDNA S plasmid was used as a vaccine in BALB/c mice. After 1-week post vaccination, elevated secretions of IFN-γ were detected in the spleens of mice after challenging them with S antigen, and the response was increased by 3–30-fold if the mouse was given a vaccination boost, indicating a specific response to S. IFN-γ producing CD4+ and CD8+ T cells were also detected in lymph nodes, spleen and lungs post immunization. However, CD8+ T cells were preferential for IFN-γ while CD4+ T cells preferentially produce IL-2. Eight weeks after immunization, T cells specific to SARS S remained in the lymph nodes, spleen and lungs [370]. Additionally, a majority of memory CD4+ and CD8+ T cells were found to be effector memory cells in lungs of mice [370]. Many memory T cells also expressed IL-7Rα, which plays a role in managing the homeostasis of memory CD8+ T cells. In another study, pcDNA vaccines of the SARS-CoV-1 structural proteins S, M and N revealed that S can induce a stronger and more lasting humoral immunity compared to the other structural proteins tested.

The researchers separated S into overlapping C- and N-terminal subunits, denoted as pcDNASa and pcDNASb in a 1:1 ratio in BALB/c mice. While the humoral immunity was strongest for the S vaccine, it induced a weaker cytotoxic T cell response in comparison to M and N pcDNA [369]. Additionally, purified lymphocytes from the pcDNASa-pcDNASb vaccinated BALB/c mice hardly proliferated when restimulated with S protein [369]. Truncated S cDNA is also effective in mounting immunity. Either deleting the TM domain (SΔTM) or the cytoplasmic domain (SΔCD) produced an effective T cell response with neutralizing antibodies in BALB/c mice [371]. Surprisingly, the role of T cells in providing immune protection appears minor. Depleting T cells from the spleen and liver of vaccinated mice still resulted in protection. On the other hand, infecting mice with SARS, and then donating T cells from vaccinated mice, did not prevent infection, although donor IgG antisera did [371]. Despite this, the T cells generated produced either IFN-γ or TFN-α in response to S antigen, indicative of S specificity. All mice vaccinated with any of the cDNA vaccines were protected from SARS infection 30 days after immunization [371]. Finally, cDNA vaccination against S can be strengthened with pcDNA-IL-2 as an adjuvant [372]. Mice vaccinated with pcDNA-S + pcDNA-IL-2 had the strongest conferred cellular and humoral immunity.

In the same study, different vaccination methods were compared using injection, oral administration and electroporation. The authors noted the preference of IgG subclasses in the different vaccines tested. 10 days after immunization, the IgG1 subclass was detected primarily in pcDNA-S + pcDNA-IL-2, while pcDNA-S vaccinated mice produced primarily IgG2α [372]. Conventional intramuscular immunization produced a better antigen-specific T cell response than electroporation, but electroporation produced better humoral immunity. Additionally, specific subsets of cytokine secreting CD4+ T cells, Th1 and Th2, were discerned and measured. In all groups, Th1(IFN-γ secreting) and Th2 (IL-4 secreting) were present, but Th1 composed the majority of Th cells, consistent with an inflammatory response associated with SARS. The addition of IL-2 as an adjuvant indicates that immunization against structural proteins alone is not enough to activate the immune system to its greatest potential. IL-2 is a modulatory cytokine for both innate and adaptive immune cells, activating Th cells, cytotoxic T cells, B cells, macrophages and Natural Killer cells.

Overall, DNA vaccines against S are probably effective, due to the production of humoral immunity followed by a T cell response, even if an infection were to still occur, possibly from a closely related virus. Upon detecting the S antigen, Th1 cells release IFN-γ, recruiting and activating phagocytes to regions where SARS is present. IgG antibodies sequester the S protein while activated white blood cells consume the viral particles. The seeming ineffectiveness of T cells in protecting against SARS as reported by Yang et al. could be attributed to the role the T cell response has to a SARS infection. T cells were detected in patients with mild to severe infections, coupled with elevated IgG. If an abundance of cells in tissues are to be infected, CD8+ cytotoxic T cells would be required to kill cells to prevent further replication of the virus, consistent with the measured higher level CD8+ T cell response over the CD4+ T cell response in SARS survivors [373].

Meanwhile, Th1 cells may modulate and enhance secretion of IgG2a to continue sequestering viral particles, thereby increasing inflammation in and chemotaxis to infected tissues. The danger of the SARS-induced T cell response resides in the cytokine storm characteristic of severe infections. In combination with the innate immune response (complement) to SARS infection, and the inflammatory response due to the virus’ pathogenicity, an imbalance of Th1 and Th2 cells could be a major contributor to the disease progression.

These vaccination studies revealed a preference for IFN-γ and IL-2 secreting CD4+ T cells, indicators of Th1 cells [370,372]. Excessive accumulation of IFN-γ in the host without the anti-inflammatory secretion of Th2 could result in hyperinflammation, overactivity of white blood cells and extreme pulmonary tissue damage in mid-late infection. Hence, it would be crucial to also mount a Th2 response when developing a vaccine.

Some SARS patients had elevated Th1 cytokines, IFN-γ, IL-1β, IL-6, and IL-12, with limited to elevated anti-inflammatory Th2 IL-10 in the blood plasma [368,374]. While additional data are needed, similar results were obtained for SARS-CoV-2 patients, indicating a preference for Th1 over Th2 cells [375]. On the other hand, prolonged overproduction of Th2 IL-10 along with elevated CD8+ T cells was associated with fatal infections [368,376], suggesting that an imbalance towards Th2 may also be lethal. Some SARS-CoV-2 patients with worsening disease displayed elevated IL-10 with decreased CD4+ and CD8+ T cells [377], suggesting that IL-10 may be secreted by monocytes rather than Th2 cells, and that the T cell suppressing role from overexpressed IL-10 is detrimental. Regardless, the balance of secreted cytokines can be easily disturbed, but it is crucial for the determination of severe disease progression in both SARS and SARS-CoV-2. While additional information is necessary to determine the nature of the T cell response during SARS-CoV-2 infection, vaccines should be able to mount a full immune response.

### 18.2. N-Protein as a Protective Antigen

The N-protein has been considered by several groups for use in vaccine design. For example, Yong et al. described in 2019 recent advances in the development of vaccines against the MERS coronavirus, and N is one of several viral structural proteins used in this endeavor, others being the S, E and nsp16 CoV proteins. These authors emphasized immune responses and potential antibody-dependent enhancement of infection, but they also discussed animal models to evaluate vaccine candidates. In another study, Jiang et al., 2020 [378], using a SARS-CoV-2 proteomic microarray, characterized the IgG and IgM antibody responses to sera from 29 convalescent Covid-19 patients to most of the viral proteins. All patients produced antibodies most abundantly to the N and S1 proteins. Moreover, Basu and Brown [379] and Lee and Koohy [380] analyzed immunogenic peptides from nucleocapsid and surface proteins of several CoVs, identifying areas of the N-proteins that are conserved and therefore of interest for vaccine development. Ahmed et al. [381] conducted similar analyses, finding regions in the N-protein that were identical between CoV-1 and CoV-2 and therefore would likely prove appropriate for cross reactive vaccine development. It is encouraging that memory T-cell responses targeting SARS-CoV-1 persisted up to eleven years post-infection [382]. Additionally, several novel approaches are now being used, such as reverse vaccinology and machine learning, to develop a vaccine against CoV-2 [383].

### 18.3. M-Protein as a Protective Antigen

The M-proteins of several coronaviruses have been shown to act as dominant protective immunogens, being antigens for the humoral response [384,385]. Specifically, the N-terminal transmembrane region of M contains a T-cell epitope cluster, and this provides a major fraction of the immunogenicity of the virus [386]. The M-protein therefore serves as one possible candidate for the development of a vaccine against one or several of the human respiratory coronaviruses. However, early studies with AIBV suggested that the M glycoprotein elicited antibodies in low titers and of limited cross-reactivity. Moreover, immunization of chickens with the purified M protein did not induce protection against virulent challenge [387,388]. However, in the same year, Saif [389] reported that the M proteins of several animal coronaviruses can induce antibodies that neutralize the viruses in the presence of complement.

Twelve years later, Okada et al. [390] showed that M DNA from SARS-CoV-1, using the pcDNA 3.1(+) plasmid vector, evoked T cell immune responses (CTL induction and proliferation) in mice against this M protein. These observations were confirmed and extended by Liu et al. [385] who showed that the M-protein of SARS-CoV-1 acts as a dominant immunogen as revealed by a clustering region of novel functionally and structurally defined cytotoxic T-lymphocyte epitopes. Soon thereafter, Zhang et al. [391] concluded that a conserved linear B-cell epitope was present in the M-protein of PEDV, and Yan et al. [392] identified a similar epitope in this M-protein. Similar developments were reported by Takano et al. [393] for the Feline Infectious Peritonitis Virus (FIPV). Immune responses to pcDNA vaccines against M protein elicited stronger lymphocyte proliferation and cytotoxic T cell lysis activity than pcDNASa and pcDNASb by week 12 post vaccination. Humoral immune responses followed an interesting trend, with M-specific antibodies reaching higher levels than pcDNASa-pcDNASb and pcDNAN within 6 weeks, but they rapidly declined over the following weeks, while pcDNASa-pcDNASb retained stable levels after week 8. These results give hope that vaccines directed against the M-proteins of human pathogenic coronaviruses, including that of SARS-CoV-2, will be forthcoming in the future.

## 19. Conclusions

The novel SARS-CoV-2 virus is projected to remain a threat to global public health for at least another two years since its first occurrence in late 2019. While vaccine and antiviral research is underway, the rapid spread and fatality due to the virus indicates that pharmaceuticals will not be enough to stop this disease. Global health policy and coordination between local and national governments will be essential in order to slow the spread. The effectivity of a future vaccine must be coupled with proper social distancing, public health practices and education [394]. In this review, we detail the extensive coronavirus genome, its proteins, and their roles in viral replication and pathogenesis. The virus is notoriously capable of evading host innate immune systems, while still inducing severe disease and inflammation. Long term adaptive immunity against the virus remains in question, placing a larger pressure on effective vaccine research [395]. Optimistically, there are many targets within the coronavirus proteome, especially in the transmembrane proteins, to counteract the severe inflammation in the interest of antivirals detailed in this review. Research efforts over the past 20 years have revealed several targetable sites in most CoV proteins with identified immunoregulatory functions.

Novel human infecting pathogenic viruses resulting from zoonotic jumping are not uncommon as virologists and epidemiologists fervently study pig and bird influenzas that may have jumped to humans. Other viruses that have caused widespread epidemics that sparked a wave of research into antivirals are HIV and Ebola [396,397]. Until recently, coronaviruses have gone largely underrepresented as growing threats to civilization, despite being responsible for two epidemics in China 2003 (SARS) and the Middle East 2012-present (MERS). Additionally, minor outbreaks of HKU1-CoV occurred in Hong Kong and the USA causing mild to severe pneumonia [398,399]. Coronaviruses are also a costly agricultural nuisance with Porcine Epidemic Diarrhea Virus and Infectious Bovine Virus, largely affecting pork and cattle supply and economy, respectively [400,401]. Despite this history, antivirals against CoVs are virtually nonexistent, making humanity pharmaceutically defenseless against the Covid-19 pandemic. Since CoVs have repeatedly challenged civilization for the past 20 years, and with the widespread dispersion of SARS-CoV-2 infections, we can expect CoVs to be a major contributor to future diseases along with influenza and antibiotic resistant bacteria. The capacitance for CoVs to genetically recombine and zoonotically jump also reveals a growing vulnerability in disease prevention strategies, especially in rural areas. Increasing globalization and rapid development into rural areas allows infectious diseases to spread into cities and other countries, leading more easily to pandemics [402].

Predicting the next epidemic disease is computationally impractical and requires massive surveillance. Nevertheless, hindering acquisition of infectious diseases can be achieved through widespread education and distribution of sanitary equipment. Such practices were performed in Africa to limit the spread of diarrheal diseases and attenuate the spread of Ebola [403,404]. In the meantime, research for antivirals and a vaccine against the SARS-CoV-2 may lead to solutions against the Covid-19 pandemic, and possibly future CoVs.

## Figures and Tables

**Figure 1 ijms-22-01308-f001:**
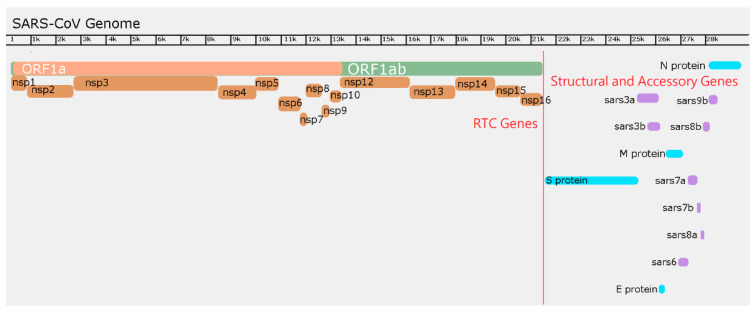
Genome graphic of SARS-CoV, showing where the genes encoding the recognized viral proteins reside.

**Figure 2 ijms-22-01308-f002:**
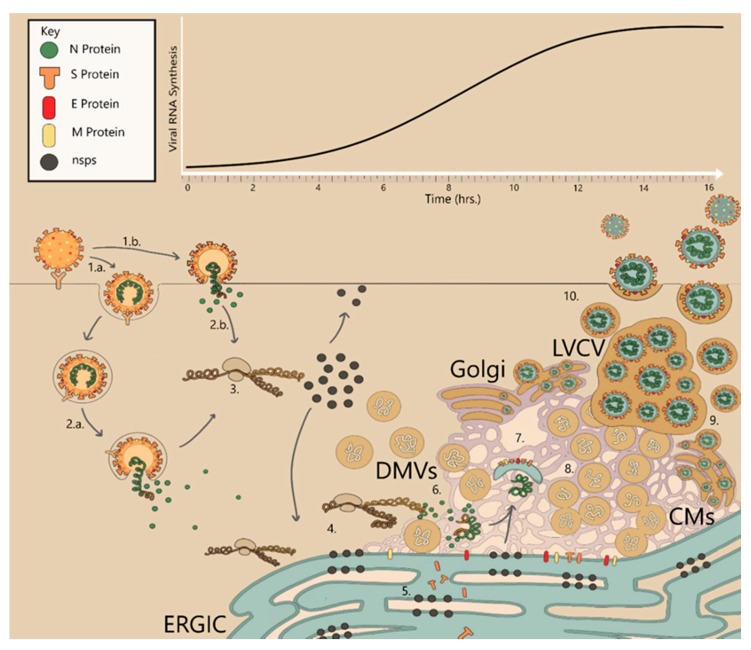
The infection cycle of coronaviruses. 1.a. Endocytic entry of virus particles after attaching to receptor proteins. 1.b. Immediate fusion at the surface of the cell upon attachment to receptor protein. 2.a. Endocytic fusion of viral and host membranes after endosomal maturation. 3. After shedding N protein from the genome, viral RNA is translated, expressing nsps from pp1a and pp1ab. Nsps migrate to the ER and perinuclear spaces, and RdRp complexes form to reverse transcribe additional viral transcripts and genomes. 4. Nsp3, nsp4 and nsp6 begin membrane rearrangements and form CMs and DMVs. 5. Viral structural proteins are produced and post-translationally modified in the ER/Golgi. 6. Viral genomes are encapsulated by N proteins. 7. Encapsulated genomes are wrapped by lipid envelopes assembled by structural proteins S, M and E. 8. Late-stage membrane rearrangements, many interconnected DMVs spread through perinuclear space with dsRNA within them. The Golgi has many virions budding from it with LVCVs attached to its membranes. 9. Mature CoV virions bud from ERGIC. 10. Exit of progeny CoV viruses via lysosomal exocytosis.

**Figure 3 ijms-22-01308-f003:**
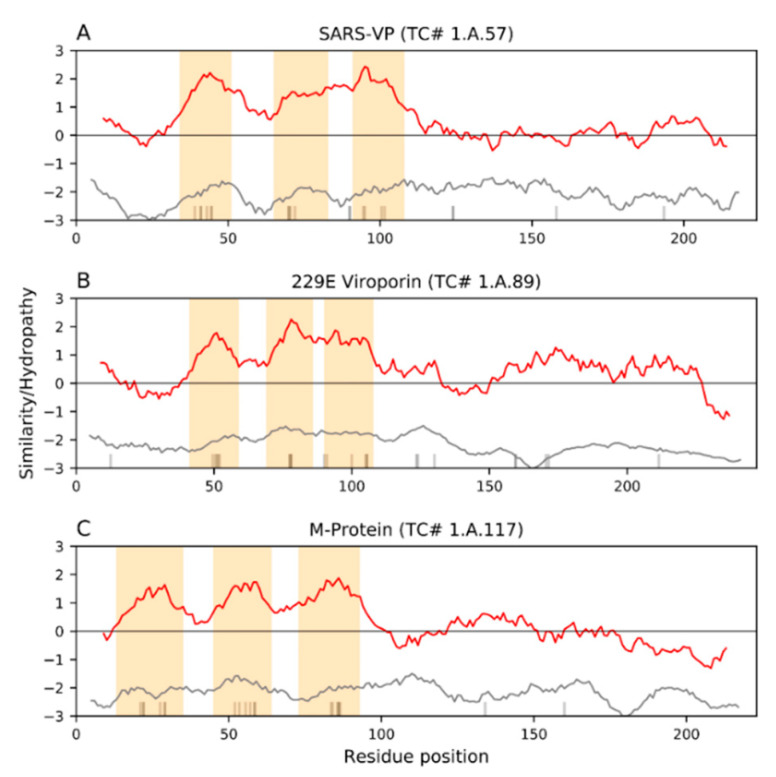
Average hydropathy and similarity within three families. (**A**) Family of SARS-3a cation-selective viroporins. (**B**) Family of HCoV-229E-4a cation-selective viroporins. (**C**). Family of M (matrix)-proteins. Red curves indicate average hydropathy, gray curves indicate average similarity per position, and vertical thin black bars on the *x*-axis indicate regions predicted to be part of TMSs. Conserved hydrophobic peaks (inferred TMSs) are highlighted with moccasin-colored bars. Proteins within each family were aligned with MAFFT [246] using the L-INS-i algorithm and then edited with trimAL [247] to keep positions with less than 30% gaps. Plots were generated with the program AveHAS [248]. Notice the high topological similarity among the three families, despite their poor sequence similarity.

**Figure 4 ijms-22-01308-f004:**
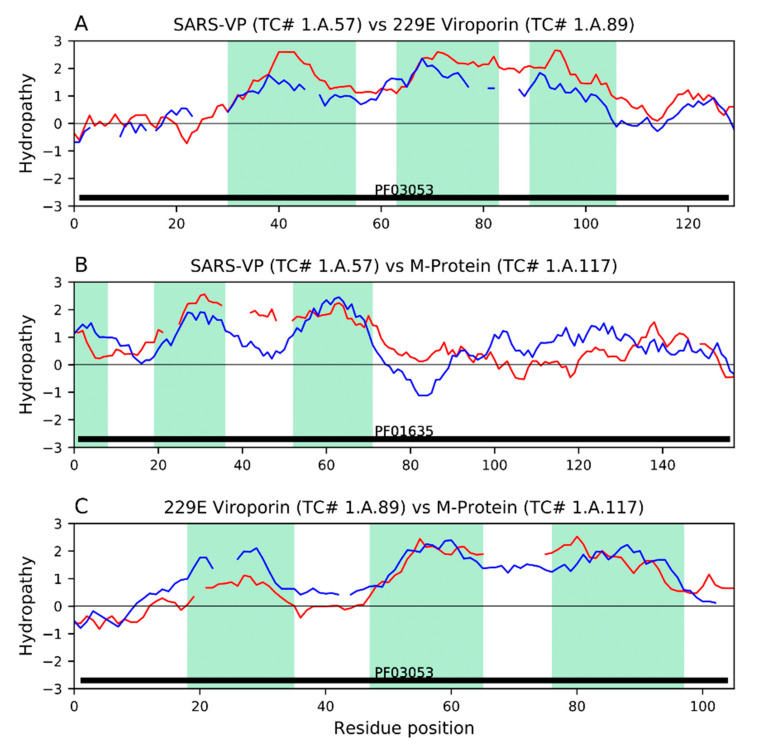
Topological relationships between (A) SARS-VP (3a) viroporins, 229E (4a) viroporins, and (C) M-proteins. Families were compared using our methodological pipeline based on the transitivity property of homology [249,250]. Hydrophobic peaks (inferred TMSs) are highlighted as green bars. Pfam domains were projected with the program GetDomainTopology [250] and drawn as solid black bars above the *x*-axis. (**A**) Hydropathy plots of representative alignments (E-value: 1.3 × 10^−5^) between a SARS-VP (3a) viroporin homolog AWV67041 (red) and a 229E (4a) viroporin homolog ADX59489 (blue). The characteristic Pfam domain of family 229E viroporins (PF03053) was projected to the SARS-VP homolog ADX59489 (E-value: 8.7 × 10^−4^). (**B**) Hydropathy plots of the representative alignments (E-value: 6.1 × 10^−7^) between a SARS-VP (3a) homolog ADX59475 (red) and an M-protein homolog ARI44791 (blue). The characteristic Pfam domain of the M-protein family (PF01635) was projected to the SARS-VP homolog ADX59475 (E-value: 4.8 × 10^−3^). (**C**) Hydropathy plots of the representative alignments (E-value: 1.4 × 10^−6^) between a 229E (3a) viroporin homolog ABQ57217 (red) and an M-Protein homolog YP_003858587 (blue). The characteristic Pfam domain of family 229E viroporin (PF03053) was projected to the M-protein homolog YP_003858587 (E-value: 1.4 × 10^−3^). Notice how the projected domains cover the entire length of the alignments in panels A-C. Altogether, the compatibility of TMS topologies (Figure 3) and the similarity of sequence characteristics between these three families suggest that they form a superfamily.

**Figure 5 ijms-22-01308-f005:**
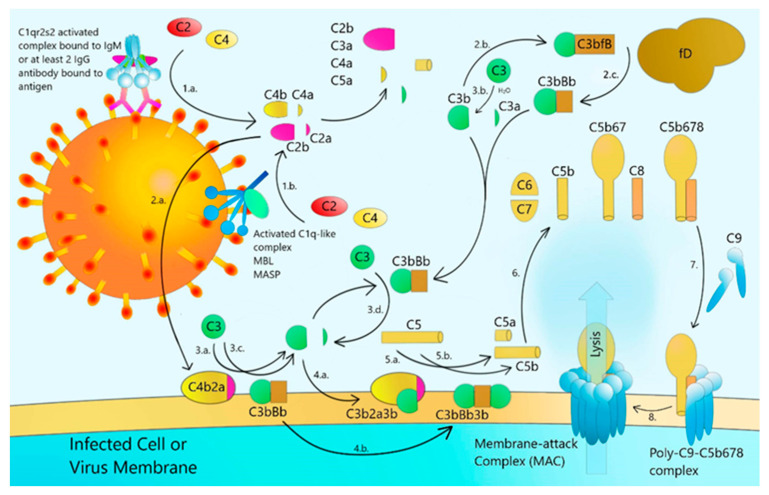
Schematic diagram of how coronavirus stimulates the complement cascade. 1.a. C1qr2s2 bound to IgM or at least 2 IgG antibodies binds to antigen. C1qr2s2 is activated and cleaves C2 and C4 into C2a, C2b, C4a, and C4b. 1.b. Activated C1q-like complex bound to MBL-MASP binds to mannose-glycosylated antigen and cleaves C2 and C4 into C2a, C2b, C4a, and C4b. 2.a. C4b and C2a bind to form C4b2a C3 convertase. 2.b. C3b binds to fB. 2.c. C3bfB is cleaved into C3bBb. 3.a. C3 is cleaved into C3a and C3b by C4b2a. 3.b. C3 is hydrolytically cleaved spontaneously into C3a and C3b. 3.c. and 3.d. C3 is alternatively cleaved by C3bBb either freely or on membrane surface. 4. C3b binds to C4b2a to form C5 convertase C4b2a3b. 4.b. C3b binds to C3Bb to form C3bBbC3b alternative C5 convertase. 5.a. C4b2a3b cleaves C5 into C5a and C5b. 5.b. C3bBbC3b cleaves C5 into C5a and C5b. 6. C5b binds to C6, C7, and C8 to form C5b678. 7. C9 is recruited to antigen presenting membranes to form poly-C9 and binds to C5b678 8. MAC is formed, and lysis occurs.

**Table 1 ijms-22-01308-t001:** Functions and properties of corona virus structural (S, E, M, N) and non-structural proteins (Nsps1–16) as well as ORFs 3a & b, 6a, 7a & b, 8a & b, and 9b.

Nsp1	Likely induces cell arrest in the G_0_/G_1_ phase and interferes with type I IFNs [70].	Nsp15	Endoribonuclease activity; Immune evasion; degrades viral polyuridine sequences to prevent host antiviral detection [71].
Nsp2	Potential nonessential role in pathogenesis [72].	Nsp16	2′O-methyl-transferase activity [73].
Nsp3	Membrane rearrangements for replication organelle formation. Viral proteolytic activity. Membrane anchoring of other viral proteins to perinuclear membranes; potential role in genome packaging. IFN antagonism.	S	Spike protein for binding/fusion/entry. Role in ER stress and syncytium formation.
Nsp4	Membrane rearrangements for replication organelle formation.	E	Envelope protein essential for virion formation and exit. Viroporin and membrane rearranging activity. Role in NLRP3 inflammasome activation.
Nsp5	Viral proteolytic activity.	M	Membrane/Matrix protein essential for virion formation. Binds to S, E and lipids to form the virial envelope-protein capsule.
Nsp6	Essential for membrane rearrangements for SARS-CoV. May induce autophagy.	N	Nucleocapsid protein, binds to viral RNAs. Necessary for packaging genome and protection from host RNAases.
Nsp7	Cofactor for RdRp complex [74].	(ORF)3a	Viroporin activity similar to E. Interacts with S, E, and M. Activates NLRP3 inflammasomes [75,76].
Nsp8	RNA binding, RNA polymerase activity and essential RdRp complex cofactor protein [74].	(ORF)3b	Putative function in upregulating cytokine secretion [75].
Nsp9	Novel ssRNA binding protein. May participate in RNA processing [77].	(ORF)6a	Colocalizes with nsp3, nsp8 and RO-associated membranes. Upstream and downstream Type I IFN antagonist [75].
Nsp10	Replicative cofactor to nsp14 [78].	(ORF)7a	Interacts with structural proteins M, E and S. May form a complex with 3a. Possibly essential for viral replication [75].
Nsp11	Unknown.	(ORF)7b	Possibly essential for viral replication. [75].
Nsp12	RNA-dependent RNA polymerase (RdRp) [79].	(ORF)8a	Viroporin activity similar to 3a and E and activates NLRP3 inflammasome [80].
Nsp13	Cofactor for the RdRp complex. Viral helicase. Unwinding duplex RNA [81].	(ORF)8b	Contributor to lysosomal stress, autophagy and inflammation; activation of NLRP3 inflammasome [82].
Nsp14	S-adenosyl methionine-dependent (N7-guanine)-methyl transferase, assembling cap1 structure at 5′ end of viral mRNA to promote translation and avoid antiviral detection. Proofreading of viral RNA transcripts [78].	(ORF)9b	Suppresses innate immunity by usurping poly-C-binding protein 2 and HECT domain E3 ligase AIP4 to degrade MAVS/TRAF3/TRAF6 signalosome [83].

**Table 2 ijms-22-01308-t002:** Membrane structures found in various coronaviruses. Bold checkmarks indicate dominating membrane structures.

Membrane Structures	SARS	MERS	MHV	HCoV-229E	PEDV	PDCov	IBV
**DMV**	✓	✓	✓	✓	✓	✓	✓
**DMS**	✓	✓	✓	✓		**✓**	**✓**
**CM**	**✓**	**✓**	**✓**	✓	✓		
**VP**	**✓**	**✓**	**✓**				
**Zippered ER**	✓	✓	✓	✓		**✓**	**✓**
**LVCV**	✓	✓	✓	✓	✓	✓	
**GVP**	✓	✓	✓				
**TB**	✓	✓	✓		✓		
**Interconnections**	**✓**	**✓**	**✓**	✓	✓	✓ (Perinuclear DMVs)	✓ (Perinuclear DMVs)

**Table 3 ijms-22-01308-t003:** Potential Functions of Coronavirus Matrix (M) proteins.

	Association with Itself and All Other Structural Proteins to Assemble Virions
1.	Interference with the host immunological response by interferon (IFN) antagonism
2.	Involvement of M in host cell cycle arrest
3.	Induction of endoplasmic reticulum (ER) stress and the unfolded protein response (UPR)
4.	Coronavirus-induced autophagy and abortive apoptosis
5.	Functioning of M as a protective antigen
6.	Viroporin activities: The E and 3a proteins, and the ability of M to substitute for E

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
