# Peer review of "The SARS-Coronavirus Infection Cycle: A Survey of Viral Membrane Proteins, Their Functional Interactions and Pathogenesis"

_ijms, 2021, doi:10.3390/ijms22031308_

Round 1

Reviewer 1 Report

Firstly, I commend the authors for a valuable review of SARS-CoV2, the causative virus for Covid-19 - a pandemic that has bothered and crippled many lives and Governments in this scientifically/technologically advanced era.

The review is very well compiled and legible. However, at times I detected it of being either verbose or redundant for certain parts. Concision of those parts will make it an even better read, reaching a broader non-expert readership.

Author Response

Thank you for taking the time to review our review.

We acknowledge that at times the terminology and explanations can be quite convoluted and saturated with terminology.  We agree that the immunology pathways and complicated experimental methods can be difficult to understand; however we must also truncate the material in order to conserve space, but we do attempt to provide our own commentary and encompassing information to soften the material.  Other than these key places, we cannot identify places where the paper is verbose.  The redundancy was incorporated so that a reader may read any section of the paper without necessarily having to read the paper from the beginning since it is quite long; however we have noticed the table of contents was removed from the paper.

Reviewer 2 Report

This aims to be a fairly comprehensive review of coronavirus molecular biology including pathogenesis, treatment and vaccination, with a special focus on some of the membrane-associated proteins.  It’s quite large in size and scope, generally detailed and well-referenced, and figures 2 and 5 seem quite useful for talks and teaching about this virus.

The review seems to go considerably beyond the scope of the membrane proteins - rather than recommending to cut out any of the sections, what about a different title that better reflects the scope of the manuscript?

……………………………..

Some rather fussy particulars of nomenclature:

The genus should have the word fully written out as Betacoronavirus (in italics) rather than the Greek symbol, or if you need to use the symbol to conserve space in a figure, for example, it should be spelled out in the legend.  It doesn’t have an official abbreviated version.  Informally referring to betacoronaviruses without capital or italics is OK.

SARS-CoV-2 is the name of this example of the species Severe acute respiratory syndrome-related coronavirus (again in italics).  Here again, the name SARS-CoV-2 shouldn’t have any other variants if we are talking about formal taxonomy.  Since this is below species rank, it does not need to be italicized. 

56-58 The CDC’s latest updated page has rather lower numbers, all lower than threefold for these.  Perhaps just remove the term four-fold?

75 I don’t know that we need the word Western here, when speaking to the audience of this journal. 

89 I think only HCoV-HKU1 has 6 ORFs – the rest have 7 or more.  Maybe just take out the word typical or rephrase?

Fig. 1 – the term subgenomic genes seems like an oxymoron, though they are translated from subgenomic RNA species.  Typically these are called structural (SEMN) and accessory (the rest) genes.

119 HIV doesn’t target CD40, and even CD4 is just a coreceptor – the real entry receptor is CCR5 if I remember correctly.

136 – the list is even longer now – there was a paper not too long ago that showed several other TMPRSS proteases could activate the SARS-CoV-2 spike.

Table 1 – your poly-C got turned into poly-copyright

291 and table 2 – we need more than HCoV (human coronavirus) to designate a particular virus – there are HKU1, OC43, 229E, NL63 in addition to the three highly pathogenic human coronaviruses, and these are all quite different from each other.

545 – You need a reference for RNA binding by nsp3 – this one covers several parts (https://www.ncbi.nlm.nih.gov/pmc/articles/PMC2395186/), or some of the individual domain structure papers also look at RNA binding. 

551-2 – I suppose the pore does not have catalytic activity, but several parts of nsp3 like the ADRP/macrodomain and papain-like protease(s) certainly do – perhaps reword this.

812-841 – There is a new paper out all about coronavirus egress – it really updates the material that is presented here:  https://www.cell.com/cell/pdf/S0092-8674(20)31446-X.pdf

Somewhere in talking about ACE2 and TMPRSS2, it would be good to mention that they appear to interact, and so would presumably colocalize on the lipid rafts.  https://www.ncbi.nlm.nih.gov/pmc/articles/PMC3020023/

1022 Neuman with a “u” rather than a “w”

1093 – better to write out HCoV-229E rather than just the latter part of the abbreviation. 

Fig. 3 – similarity to what?  And it seems like there is considerable overlap in the content of Figs. 3 and 4.  Perhaps these two could be consolidated into a single figure somehow? 

Author Response

Thank you for your extensive recommendations.

We have corrected the nomenclature.

56-58 - removed the word fourfold

75 - Taken out the word western

89 - Isn't saying at least 6 ORFs implying that their could be more than 6 ORFs?  We have left this the same.

Fig 1 - Now called Structural and Accessory Genes

119 - HIV receptor corrected to CD4, CCR5 is the Co-receptor

136 - we have extended this list, thank you for educating us on this.

Table 1 - comical error fixed 

Table 2 and HCoV turned to HVoV-229E

545 - reference added for nsp3 nucleotide binding

551 - made it more clear that the pore does not have confirmed catalytic activity, but that there are several involved nsps that do have catalytic activity.

812 - included!  CoV egress information is an exciting find!

Included possible lipid raft localization of TMPRSS2 and ACE2 and promotion of syncitia formation in respiratory epethelial tissue.

1022 Neuman

1093 Fixed, this was a typo

Fig 3, we will keep it as two separate figures, but make the information more clear in our comparative analysis.

Again, thank you for taking the time to edit our extensive review.